# Addressing Instrument-Outcome Confounding in Mendelian Randomization through Representation Learning

**Shimeng Huang** [1]   **Matthew R. Robinson** [1]   **Francesco Locatello** [1]

## Abstract

Mendelian Randomization (MR) is a prominent observational epidemiological research method, designed to address unobserved confounding when estimating causal effects. It is closely related to instrumental variable (IV) methods, where genetic variants serve as instruments to infer causal relationships from observational data. However, the core assumptions required for valid IV analysis—particularly the independence between instruments and unobserved confounders—are untestable and often violated in practice. In MR, such violations commonly arise when genetic variants are correlated with environmental factors (e.g., population stratification and assortive mating), leading to confounding between instruments and outcomes. At the same time, MR studies increasingly include data collected across multiple environments or populations, providing an opportunity to address these violations. Leveraging this setting, we propose a representation learning framework that exploits multi-environment data to recover latent exogenous components of genetic instruments suitable for causal inference. We provide theoretical insights into when and how the learned components can act as valid instruments, and we demonstrate the effectiveness of our approach through simulations and semi-synthetic experiments using genetic data from the All of Us Biobank.

## 1. Introduction

Understanding cause-and-effect relationships between risk factors and disease outcomes is central to epidemiology and genetic research. When randomized controlled trials are infeasible or unethical, researchers must rely on observational data, where unobserved confounding poses a major challenge. Mendelian Randomization (MR), an instrumental variable (IV) approach (Angrist et al., 1996; Anderson & Rubin, 1949) that uses genetic variants as instruments, has therefore become a prominent tool for causal inference in genetic epidemiology (Sanderson et al., 2022).

Valid IV estimation requires instruments that satisfy relevance, exchangeability, and exclusion restriction (see Definition 2.1). In MR, while relevance could be empirically assessed, exchangeability and exclusion restriction are fundamentally untestable and frequently violated in practice (Mason et al., 2025; Swerdlow et al., 2016). A primary source of such violations is population stratification, where genetic variants and phenotypes share common ancestry- or environment-driven influences, biasing causal estimates in MR studies (Sanderson et al., 2022).

Crucially, multiple large-scale biobanks such as All of Us (The All of Us Research Program Investigators, 2019) are now available that could be utilised if there were a way of conducting MR reliably across populations. Although the underlying causal variants for traits are likely to be shared across populations, the correlations (linkage disequilibrium) and allele frequencies among genetic markers vary significantly due to distinct demographic histories (Wang et al., 2020; 2023). This contrast suggests that observed genetic variants can be viewed as mixtures of invariant biological signals and environment-specific confounding components. In this work, we leverage this structure and use representation learning to exploit cross-environment invariance (Ahuja et al., 2024; Yao et al., 2025), with the goal of isolating the invariant component of confounded genetic instruments for valid causal inference.

### 1.1. Contributions and Overview

Motivated by environmental confounding and the availability of multi-environment data suitable for MR, we propose a representation learning framework for IV estimation in which both instruments and outcomes may be confounded by environment-related factors. Our approach exploits invariance across environments to identify latent components of a confounded instrument that remain stable under environmental shifts. We show that recovering this invariant component up to a transformation—consistent with typical

---

[1]Institute of Science and Technology Austria (ISTA). Correspondence to: Shimeng Huang <shimeng.huang@ist.ac.at>.

*Proceedings of the 43$^{rd}$ International Conference on Machine Learning*, Seoul, South Korea. PMLR 306, 2026. Copyright 2026 by the author(s).

guarantees in representation learning—is sufficient for valid downstream causal inference, effectively mitigating violations of the exchangeability assumption. While some recent works have explored learned representations for IV estimation (Cheng et al., 2024a;b), they generally lack theoretical guarantees on recovering the latent valid instruments.

Our contributions are as follows. To the best of our knowledge, this is the first work to apply multi-environment representation learning to address instrument-outcome confounding in MR, a setting where the exchangeability of observed genetic instruments is systematically violated and existing representation-based IV methods do not offer identifiability guarantees. We introduce a measure-theoretic notion of identifiability and connect it to existing definitions in representation learning. We adapt multi-environment identifiability theory to MR (Section 2) and establish guarantees for recovering latent valid instruments under various mixing mechanisms (Section 3). In contrast to prior identification results, we explicitly characterize the effect of misspecifying the latent dimension and show empirically that such misspecification can bias downstream causal estimates (Figure 5). We further analyze how the use of learned representations affects the validity and efficiency of downstream causal estimation (Section 4). Finally, we present synthetic and semi-synthetic experiments demonstrating effective bias correction under environmental confounding (Section 5). Proofs are deferred to Appendix D.

## 1.2. Other Related Works

**Estimation with Invalid Instruments.** There exists a significant body of work that addresses IV estimation when the instruments are invalid. Popular methods include those that assume validity holds for the majority of instruments (e.g., median-based estimators) or enforce sparsity on invalid effects (Kang et al., 2024). In the context of Mendelian Randomization, specific methods like MR-Egger (Bowden et al., 2015; Rees et al., 2017) and GENIUS (Tchetgen Tchetgen et al., 2021) allow for invalid instruments under specific parametric or structural assumptions. In contrast, our framework addresses systematic confounding in the instruments by explicitly modeling and disentangling these factors via invariance without imposing specific assumptions on the form of causal effects.

**Identifiability in Representation Learning.** Our identification strategy draws upon recent advances in causal representation learning. The challenge of identifying latent causal variables from high-dimensional observations without supervision is generally ill-posed (Hyvärinen et al., 2019; Locatello et al., 2019). However, foundational theoretical results have established that identifiability is achievable given auxiliary information, such as multiple environments (Hyvärinen & Morioka, 2016; Khemakhem et al., 2020). Building on this, a series of works have explored the iden-

tification of latent variables based on soft-intervention or multi-environment data (Zhang et al., 2023; Ahuja et al., 2023; 2024). Yao et al. (2025) provides a unifying view of different RL methods using invariance constraints. While these were initially proposed without clear applications, we discovered that the insights extend to the MR setting, specifically tailoring the identification results to separate valid instrumental components from environmental confounders.

**Representation Learning for Mendelian Randomization.** Recent work has explored deep representations of outcomes for MR (Reddy et al., 2025). In contrast, our framework applies representation learning to the *instrument* itself to remove instrument-outcome confounding.

## 2. Motivation and Problem Setup

**Notation.** *We let* $[K] := \{1, \ldots, K\}$. *For a vector* $x \in \mathbb{R}^n$, $x^{:k}$ *and* $x^{k:}$ *denote the first* $k$ *elements and remaining elements, respectively. Similarly, for* $X \in \mathbb{R}^{m \times n}$, $X^{:k}$ *denotes the first* $k$ *columns,* $X_{k:}$ *the rows after the first* $k$ *rows, and* $X_i^j$ *the* $(i, j)$*-th entry. Superscripts with brackets denote environments (e.g.,* $X^{(k)}$*). Finally,* $X \stackrel{d}{=} Y$ *(resp.* $X \stackrel{d}{\neq} Y$*) indicates equality (resp. inequality) in distribution.*

### 2.1. Existing Issues in Mendelian Randomization

There are three classic conditions on an instrumental variable (IV) that allow valid testing for the null hypothesis of no causal effect. An IV is commonly referred to as a *valid* instrument if it satisfies these conditions.

**Definition 2.1** (Valid instruments). Suppose $H$ taking values in $\mathcal{H}$ is an unobserved confounder between an exposure $D \in \mathbb{R}^d$ and a response $Y \in \mathbb{R}$. A variable $Z \in \mathcal{Z}$ is a *valid instrument* for $D$ w.r.t. $Y$ if it satisfies: (1) **Relevance:** $Z \not\perp\!\!\!\perp D$, (2) **Exchangeability:** $Z \perp\!\!\!\perp H$, and (3) **Exclusion Restriction:** $Z \perp\!\!\!\perp Y \mid H, D$.

As discussed in Section 1, in MR studies, genetic variants are often confounded with phenotypes through population structure or assortative mating, violating the exchangeability condition in Definition 2.1. An illustrative example is given below.

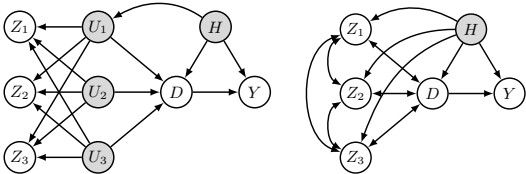

*Figure 1.* Left: DAG with disentangled latent variables that generate $Z$, where $U_2$ and $U_3$, if observed, are valid instruments for $D$ with respect to $Y$. Right: ADMG without considering the disentangled latent variables, $Z_1$ to $Z_3$ are all invalid instruments due to $Z_1$'s violation of exchangability.[1]

**Example 2.1.** *Suppose we are interested in estimating the average causal effect (ACE) of having type II diabetes (D) on developing coronary artery disease (Y) from observational data, where both are confounded by unobserved population-level factors (H) arising from ancestry and social mating patterns. Let $Z := (Z_1, Z_2, Z_3)$ be genetic variants selected as candidate instruments. Suppose each is a combination of latent variables $U := (U_1, U_2, U_3)$. Among these, $U_2$ and $U_3$ represent stochastic genetic recombination governed by Mendel's law (valid), while $U_1$ is influenced by population factors $H$ (invalid). Figure 1 (left) illustrates this. When only the mixed variants $Z$ are observed, the influence of $U_1$ induces confounding with $Y$, rendering the observed instruments invalid (Figure 1, right).*

In Example 2.1, recovering the latent variables $U$ from the observed variables $Z$ would yield two valid instruments for the treatment $D$. In the notation of the general setup described next, this amounts to identifying a valid component $W := (U_2, U_3)$ alongside an invalid component $V := U_1$ that is influenced by $H$. We formally describe the problem setup motivated by this example below.

## 2.2. Problem Setup

We consider a multi-environment[2] Mendelian Randomization (MR) setup formally described below.

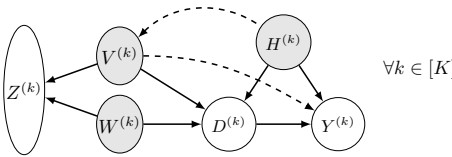

$\forall k \in [K]$

*Figure 2.* Illustration of our general setup where $Z$ is a complex, entangled instrument containing some valid information (represented by $W$) as IV for $D$ with respect to $Y$, and some invalid information (represented by $V$). $V$ and $W$ are not directly observed and are not necessarily subvectors of $Z$.

**Setting 2.1.** *For each of $K$ environments, we observe i.i.d. data $\mathcal{D}_k := \{(Z_i^{(k)}, D_i^{(k)}, Y_i^{(k)})\}_{i=1}^{n_k}$ comprising an exposure $D^{(k)} \in \mathbb{R}^d$, a response $Y^{(k)} \in \mathbb{R}$ confounded by unobserved $H^{(k)}$, and a high-dimensional candidate instrument $Z^{(k)} \in \mathcal{Z}$. We assume $Z^{(k)}$ is generated as $Z^{(k)} = f(W^{(k)}, V^{(k)})$ via an invertible mixing function $f$, where the latent components $W^{(k)} \in \mathbb{R}^p$ and $V^{(k)} \in \mathbb{R}^q$ satisfy:*

*(1) $W^{(k)}$ and $V^{(k)}$ are independent in each environment, i.e., $W^{(k)} \perp\!\!\!\perp V^{(k)}$ for all $k \in [K]$.*

*(2) For all $k \in [K]$, $W^{(k)}$ is a valid instrument whereas $V^{(k)}$ is invalid. In particular, $W^{(k)} \perp\!\!\!\perp H^{(k)}$ while*

*$V^{(k)} \not\!\perp\!\!\!\perp H^{(k)}$.*

*(3) The distribution of $W^{(k)}$ is invariant across environments, i.e., $W^{(k)} \stackrel{d}{=} W^{(k')}$ for all $k, k' \in [K]$, while the distribution of $V^{(k)}$ varies across environments driven by shifts in $H^{(k)}$.*

*Thus, while $Z^{(k)}$ is an invalid instrument due to $V^{(k)}$, there exists a function $\varphi : \mathcal{Z} \to \mathbb{R}^p$ (corresponding to the inversion of $f$ onto the $W$ coordinates) such that $\varphi(Z^{(k)})$ constitutes a valid instrument.*

Setting 2.1 describes a flexible data-generating process motivated by challenges in MR. In particular, we model the sources of variation in the genetic variants via $W$ and $V$, where $W$ captures invariant variations across environments while $V$ captures variant variations driven by environmental heterogeneity. Our goal is to disentangle $W$ from $V$ to eliminate the environmental confounding affecting the instrument $Z$. Importantly, this disentanglement does not remove the unobserved confounding between the exposure $D$ and the outcome $Y$; rather, it recovers a valid instrument $W$ so that IV methods can be correctly applied. This setting also accommodates effect heterogeneity: the functional relationships (including the causal effect of $D$ on $Y$) are permitted to vary across environments, so the causal effect can be estimated either jointly or separately for each environment, depending on the specific application.

*Remark* 2.2. The invertibility of the mixing function $f$ in Setting 2.1 is sufficient but not necessary for our purposes: if $f$ is non-injective and some information about $W$ is unrecoverable from $Z$, the recoverable component may still constitute a valid instrument.

In Setting 2.1, we assume that the latent variables $W$ and $V$ satisfy $W \perp\!\!\!\perp V$. This assumption is necessary for the following reason: if $Z$ cannot be expressed as a function of two independent components, one of which satisfies all conditions in Definition 2.1, then no transformation of $Z$ can serve as a valid instrument for $X$ with respect to $Y$. This impossibility result is formalized in Proposition 2.3. Conversely, if $Z$ can be mapped to a variable that constitutes a valid instrument, then $Z$ may be viewed as being generated from this variable and an additional independent variable.

**Proposition 2.3.** *Given an arbitrary random variable $Z \in \mathcal{Z}$, if there does not exist a function $\ell$ such that $Z = \ell(A, B)$ where $A \perp\!\!\!\perp B$ and $A$ satisfies all conditons in Defintion 2.1, then there is no function $\varphi$ such that $\varphi(Z)$ satisfies all three conditions in Defintion 2.1, and thus the ACE is not identifiable given $Z$, $D$, and $Y$.*

Theoretical results in representation learning show that, under general conditions, one can recover the latent variables only up to a bijective transformation (see e.g., Ahuja et al., 2024). Fortunately, this is sufficient for our purpose. As shown in Proposition 2.4, a key property of valid instruments

---

[1]DAG: directly acyclic graph. ADMG: acyclic directed mixed graph.

[2]In the MR context, "environments" typically correspond to distinct populations (e.g., defined by ancestry); we use the two terms interchangeably throughout.

is that any bijective transformation of a valid instrument remains valid.

**Proposition 2.4** (Bijective transformations of a valid instrument are also valid). *If $Z$ is a valid instrument for $D$ with respect to a response $Y$ (Definition 2.1) and $\kappa$ is a bijective measurable function $\kappa : \mathcal{Z} \to \mathcal{Z}', z \mapsto \kappa(z)$. Then $\kappa(Z)$ is also a valid instrument for $D$ with respect to $Y$.*

In the rest of this section, we omit the superscript "$(k)$" for clarity, as the statements apply to any $k \in [K]$. Let $f^{-1}$ denote the inverse function of the mixing function $f$. Without loss of generality, we assume that the first $p$ coordinates of $f^{-1}(Z)$ correspond to $W$, and we denote this subvector of $f^{-1}(Z)$ as $\varphi_0(Z) \coloneqq \left(f^{-1}(Z)\right)^{:p} = W$, and the last $q$ coordinates of $f^{-1}(Z)$ correspond to $V$. Under a partially linear structural causal model (PLSCM, see Definition A.1 in Appendix A), with $\varphi_0$ and the true causal parameter $\theta_0$, it holds that

$$\mathbb{E}[\varphi_0(Z)(Y - D^\top \theta_0)] = 0. \tag{1}$$

The condition in (1) is often referred to as the *moment restriction* in IV (see e.g., Wooldridge, 2010). A sufficient and necessary condition of using (1) to identify $\theta_0$ is the following *rank condition*:

$$\mathbb{E}[\varphi_0(Z)D^\top]v = 0 \quad \implies \quad v = 0. \tag{2}$$

Under Setting 2.1 and the rank condition in (2), we have that $\varphi_0(Z)$ is a valid instrument (Definition 2.1), and if $\varphi_0$ is known, $\theta_0$ is uniquely identified by (1). When $\varphi_0$ is unknown, $\varphi_0$ and $\theta_0$ are in general not jointly identifiable based on the moment restriction alone. This is due to the following reason: If (1) holds, then for any measurable function $m$, it also holds that $\mathbb{E}[m(\varphi_0(Z))(Y - \theta_0 D)] = 0$. However, as our interests lie in testing and estimating the Average Causal Effect (ACE) of $D$ on $Y$, we do not need to identify $\varphi_0$ itself. In fact, any bijective transformation of $\varphi_0(Z)$ that still satisfies the rank condition (2) is not only a valid instrument (as shown in Proposition 2.4), but also sufficient to identify $\theta_0$ as well. Corollary 2.5 provides one sufficient condition of such transformations.

**Corollary 2.5** (Rank perserving bijective transformations of an identifying IV). *Suppose $\varphi(Z)$ which takes values in $\mathbb{R}^p$ is a valid instrument (Definition 2.1) and satisfies the rank condition (2). If $h$ is an invertible affine transformation, i.e., there exists a constant full rank matrix $B \in \mathbb{R}^{p \times p}$ and a constant vector $c \in \mathbb{R}^p$ satisfying*

$$h(\varphi(Z)) = B\varphi(Z) + c,$$

*then $h(\varphi(Z))$ is also a valid instrument and*

$$\mathbb{E}[h(\varphi(Z))(Y - D^\top \theta)] = 0$$

*identifies the causal effect $\theta_0$.*

The problem of recovering a latent component from ob-

served data is often referred to as the identification problem in representation learning. This is discussed in Section 3.

## 3. Identifying Latent Components of Confounded Instruments

We first provide a measure-theoretic definition of the identification of latent components. In Lemma 3.2 we show that it is equivalent to a functional relationship between the representation and the latent components—a notion commonly used in the representation learning literature.

**Definition 3.1** (Identification of latent components). Let $U \in \mathbb{R}^m$ be the set of latent variables that generate the observed variable $Z \in \mathcal{Z}$ via a mixing function. Let $\eta : \mathcal{Z} \to \mathbb{R}^d$ be a measurable function and suppose both $U$ and $\eta(Z)$ admit densities. For any given $S \subseteq [m]$, we say that $U^S$ is *perfectly identified* by $\eta$ if[3]

$$\sigma\left(\eta(Z)\right) = \sigma(U^S), \tag{3}$$

and we say that that $U^S$ is *partially identified* by $\eta$ if

$$\emptyset \subsetneq \sigma\left(\eta(Z)\right) \subsetneq \sigma(U^S). \tag{4}$$

**Lemma 3.2** (Functional characterization of identification). *In Definition 3.1, perfect identification is satisfied if and only if there exists a measurable bijection $\delta : supp(U^S) \to supp(\eta(Z))$ such that $\eta(Z) = \delta(U^S)$ a.s.; partial identification is satisfied if and only if there exists a measurable function $\delta$ such that $\eta(Z) = \delta(U^S)$ a.s., but there exists no measurable function $\omega$ such that $U^S = \omega(\eta(Z))$ a.s..*

If $W^{(k)}$ is a valid instrument in environment $k \in [K]$ and it is identified at least partially by $\varphi(Z^{(k)})$, then the ACE in this environment can be identified by the independence restriction (1); if $V$ is also identified at least partially, one may be able to improve the efficiency of the estimation by using $V$. We discuss this and the caveat when using the learned latent components of $Z$ in Section 4.

### 3.1. Identifying $W$ via Distributional Invariance

From Definition 3.1, we see that if $\varphi(Z)$ identifies $W$, it must preserve the information contained in $W$. While this condition cannot be enforced directly, under certain conditions, we can enforce it by preserving the information of all $Z$ using an autoencoder, that is, a pair of parameterized functions $(m_{\text{en}}, m_{\text{de}})$ such that the observable $Z$ can be (ideally) perfectly reconstructed. Concretely,

$$m_{\text{de}} \circ m_{\text{en}}(Z) = Z. \tag{5}$$

Under Setting 2.1, we have that $W^{(k)} \sim Q_\omega$ for all $k \in [K]$. We call this relationship a *distributional invariance*, which can be imposed as an invariance constraint on the autoencoder: for a given dimension $\hat{p}$ such that (without

---

[3]We interpret all $\sigma$-algebra equalities as equalities of their $P$-completions, i.e., up to null sets.

loss of generality) the first $\hat{p}$-dimensions of the encoder output serve as the representation of $W$, the distribution of this representation is required to remain stable across environments, i.e.,

$$\forall\, k, k' \in [K], \ \ m_{\text{en}}(Z^{(k)})^{:\hat{p}} \stackrel{d}{=} m_{\text{en}}(Z^{(k')})^{:\hat{p}}. \quad (6)$$

In practice, a common nonparametric approach to quantify the level of invariance between the representations of $W$ across different environments is to estimate the Maximum Mean Discrepancy (MMD, Gretton et al., 2012) using samples of the representations.

We first consider the case where the mixing function $f$ is an injective polynomial of a finite degree (Assumption 3.3). Under this assumption, if the changes in the distribution of $V$ are modular and sufficiently variable (Assumption 3.4), then $W$ can be identified up to an affine transformation.

**Assumption 3.3** (Common polynomial mixing). For all $k \in [K]$, the mixing function $f^{(k)} : \mathbb{R}^p \times \mathbb{R}^q \to \mathcal{Z}$ where $\mathcal{Z} = \mathbb{R}^{d_z}$, is an injective polynomial of degree $L$ (see Definition A.2 in Appendix A).

**Assumption 3.4** (Modular and sufficient variability under common polynomial mixing). There exists a collection $\mathcal{S} := \{S_i\}_{i \in [m]}$ such that $\forall i \in [m]$, $S_i \subseteq [q]$, and $\bigcup_i S_i = [q]$, satisfying:

(i) For each $S \in \mathcal{S}$, there exists $k_1, k_2 \in [K]$ such that $V^{S,(k_1)} \stackrel{d}{\neq} V^{S,(k_2)}$, $V^{-S,(k_1)} \stackrel{d}{=} V^{-S,(k_2)}$, and $V^{S,(k)} \perp\!\!\!\perp V^{-S,(k)}$ for $k \in \{k_1, k_2\}$.

(ii) For any $S \in \mathcal{S}$ and $k_1, k_2 \in [K]$ satisfying (i), if there exists $u \in \mathbb{R}^{|S|}$ such that $u^\top V^{S,(k_1)} \stackrel{d}{=} u^\top V^{S,(k_2)}$, then $u = \mathbf{0}$.

*Remark* 3.5. Ahuja et al. (2024) also considers common polynomial mixing functions. However, Assumption 3.4 does not rely on their restrictive assumption of an additive noise SCM over the latent variables or that interventions act solely on the exogenous noise terms. Furthermore, unlike the SCM framework where distribution shifts are constrained to propagate from independent noise terms, Assumption 3.4 allows for arbitrary changes in the joint distribution (including the dependence structure) of the block $V^S$, provided it is transiently modular.

See Example C.1 for an example where Assumption 3.4 is satisfied. Theorem 3.6 considers the identification of $W$ under polynomial mixing.

**Theorem 3.6** (Identification of $W$ under polynomial mixing). *Consider Setting 2.1 and assuming Assumption 3.3 and Assumption 3.4 hold. An autoencoder $(m_{en}, m_{de})$ where $m_{de}$ is also an injective polynomial of degree $L$ (Definition A.2), which satisfies the reconstruction identity (5) and the invariance constraint (6) with $\hat{p}$, satisfies that $m_{en}(Z)^{:\hat{p}}$ perfectly identifies $W$ if $\hat{p} \geq p$. More specifically, $m_{en}(Z)^{:\hat{p}}$ identifies $W$ up to an affine transformation. That is, there*

*exist a constant matrix $A \in \mathbb{R}^{\hat{p} \times p}$ with full column rank and a constant vector $a \in \mathbb{R}^{\hat{p}}$ such that*

$$m_{en}(Z^{(k)})^{:\hat{p}} = AW^{(k)} + a$$

*for all $k \in [K]$. Moreover, $m_{en}(Z)^{:\hat{p}}$ partially identifies $W$ if $\hat{p} < p$ (provided the invariant latent dimensions are non-degenerate). Specifically, there exist a constant matrix $A' \in \mathbb{R}^{\hat{p} \times p}$ and a constant vector $a' \in \mathbb{R}^{\hat{p}}$ such that*

$$m_{en}(Z^{(k)})^{:\hat{p}} = A'W^{(k)} + a'$$

*for all $k \in [K]$.*

Theorem 3.6 tells us that if one chooses $\hat{p} < p$, the reconstruction identity (5) and the invariance constraint (6) may not be sufficient to ensure that the information of $W$ is preserved by $m_{\text{en}}(Z)^{:\hat{p}}$, in which case perfect identification of $W$ is not guaranteed. Example C.2 illustrates this.

*Remark* 3.7. The identification result in Theorem 3.6 is similar to Theorem 2 in Ahuja et al. (2024) but under different assumptions (see Remark 3.5). Moreover, compared to Ahuja et al. (2024), we also clarify the impact of misspecifying the dimension of the latent variable $W$.

**Corollary 3.8** (Representations of $W$ identify causal effects under polynomial mixing). *For any $k \in [K]$, representation $\widehat{W}^{(k)}$ obtained by Theorem 3.6 satisfies the rank condition (2) as long as $W^{(k)}$ satisfies this condition.*

The above identification results based on polynomial mixing (Assumption 3.3) can be generalized to general diffeomorphic mixing functions (Assumption 3.9). Under this more general setting, we describe a stronger version of Assumption 3.4 regarding the degree of changes in $V^{(k)}$'s distribution, in Assumption 3.10.

**Assumption 3.9** (Diffeomorphic mixing). The mixing function $f : \mathbb{R}^{p+q} \to \mathcal{Z}$ such that $Z^{(k)} = f(W^{(k)}, V^{(k)})$ for all $k \in [K]$, is a $C^1$ diffeomorphism.

**Assumption 3.10** (Sufficient variability under general diffeomorphic mixing). The distribution of $V^{(k)}$ satisfies Assumption 3.4 condition (i) and the following condition (ii):

(ii) Given $S \in \mathcal{S}$ and environments $k_1, k_2$ satisfying Condition (i), for any smooth function $h : \mathbb{R}^{p+q} \to \mathbb{R}^{\hat{p}}$, if the distribution of the output is invariant, i.e.,

$$h(W^{(k_1)}, V^{(k_1)}) \stackrel{d}{=} h(W^{(k_2)}, V^{(k_2)}),$$

then $h$ must be locally constant with respect to $S$, i.e.,

$$\frac{\partial h}{\partial v_j}(w, v) = \mathbf{0} \quad \forall j \in S, \forall (w, v) \in \mathbb{R}^p \times \mathbb{R}^q.$$

The following theorem states how $W$ can be identified given general diffeomorphic mixing using diffeomorphic encoders. This can be achieved by neural networks where diffeomorphic functions are enforced or by specific networks such as normalizing flows (Kobyzev et al., 2020; Rezende & Mo-

hamed, 2015), whose invertibility and exact marginal likelihoods have also been leveraged for latent-variable recovery (Li et al., 2020). We provide two additional identification results under more general assumptions in Appendix B.

**Theorem 3.11** (Identification of $W$ given general diffeomorphic mixings). *Consider Setting 2.1 and suppose Assumptions 3.9 and 3.10 hold. Assume further that $W^{(k)} \perp\!\!\!\perp V^{(k)}$ for all $k \in [K]$ and $W^{(k)}$ is invariant across environments. Let $\mathcal{A}(\hat{p}, \hat{q})$ denote the class of autoencoders $(m_{\mathrm{en}}, m_{\mathrm{de}})$ whose encoder $m_{\mathrm{en}} : \mathcal{Z} \to \mathbb{R}^{\hat{p}+\hat{q}}$ is a diffeomorphism and whose decoder $m_{\mathrm{de}}$ is a smooth function. If an encoder $m_{\mathrm{en}} \in \mathcal{A}(\hat{p}, \hat{q})$ satisfies the invariance constraint (6) with its first $\hat{p}$ dimensions, then the first $\hat{p}$ components of the latent representation depend only on $W$. Specifically, there exists a smooth function $\psi : \mathbb{R}^p \to \mathbb{R}^{\hat{p}}$ such that:*

$$m_{\mathrm{en}}(Z)^{:\hat{p}} = \psi(W) \quad \textit{almost surely.}$$

*If $\hat{p} \geq p$, $W$ is identified up to a diffeomorphism ($\psi$ is an embedding). If $\hat{p} < p$, $W$ is partially identified ($\psi$ is a projection).*

### 3.2. Identification of $V$

The identification of the non-invariant component $V$ is, in general, not achievable without further constraints (also pointed out by Yao et al. (2025)). In fact, even if $W$ is perfectly identified by $m_{\mathrm{en}}(Z)^{:p}$ under the reconstruction identity and the distributional invariance constraints, the complement $m_{\mathrm{en}}(Z)^{p:}$ need not identify $V$. An example is given in Example C.3.

Nevertheless, under the same assumptions as in Theorem 3.6 (polynomial mixing), if the autoencoder satisfies an additional independence constraint, $V$ is also identified. This is formally stated in Corollary 3.12.

**Corollary 3.12** (Identification of $W$ and $V$ under common polynomial mixing). *Assume Setting 2.1 holds and $W$ has a finite second moment with a positive definite covariance matrix. Suppose Assumptions 3.3 and 3.4 hold. If an autoencoder $(m_{en}, m_{de})$ with $m_{de}$ being an injective polynomial satisfying the conditions of Theorem 3.6 (with $\hat{p} \geq p$) additionally satisfies the* independence constraint*:*

$$m_{en}(Z^{(k)})^{:\hat{p}} \perp\!\!\!\perp m_{en}(Z^{(k)})^{\hat{p}:}, \tag{7}$$

*then it identifies both $W^{(k)}$ and $V^{(k)}$ up to affine transformations. Specifically:*

$$m_{en}(Z^{(k)})^{:\hat{p}} = AW^{(k)} + a, \;\; m_{en}(Z^{(k)})^{\hat{p}:} = BV^{(k)} + b,$$

*where $A$ and $B$ are constant matrices with full column rank.*

In the case of general diffeomorphic mixing, satisfying the additional independence constraint is not sufficient to identify $V$. An example is provided in Example C.4.

### 3.3. Identification in Practice

In this section, we describe how identification of the latent components of the observed instruments can be achieved in practice given multi-environment data.

Following the theory developed in Section 3.1 and 3.2, the practical implementation is to train an autoencoder subject to the invariance constraint to identify the invariant component $W$, and additionally an independence constraint to identify $V$. To prevent the representations to collapse during training, we add a loss term to discourage diminishing determinant of the learned representations (see Appendix E). Given a set of observed instruments $\{Z^{(k)}\}_{k=1}^{K}$, the autoencoder is trained by minimizing a weighted sum of losses:

$$(\hat{m}_{\mathrm{en}}, \hat{m}_{\mathrm{de}}) := \underset{(m_{\mathrm{en}}, m_{\mathrm{de}})}{\arg\min} \; \mathcal{L}_{\mathrm{rec}} + \lambda_1 \mathcal{L}_{\mathrm{inv}} + \lambda_2 \mathcal{L}_{\mathrm{ind}}, \tag{8}$$

where $\mathcal{L}_{\mathrm{rec}}$ and $\mathcal{L}_{\mathrm{inv}}$ are reconstruction, invariance, and independence losses, respectively, defined as

$$\mathcal{L}_{\mathrm{rec}} := \sum_{k \in [K]} \mathbb{E}\left[\left(m_{\mathrm{de}} \circ m_{\mathrm{en}}(Z^{(k)}) - Z^{(k)}\right)^2\right],$$

$$\mathcal{L}_{\mathrm{inv}} := \sum_{\substack{j,k \in [K] \\ j \neq k}} \mathrm{Inv}\left(m_{\mathrm{en}}(Z^{(j)})^{:\hat{p}}, m_{\mathrm{en}}(Z^{(k)})^{:\hat{p}}\right),$$

$$\mathcal{L}_{\mathrm{ind}} := \sum_{k \in [K]} \mathrm{Ind}\left(m_{\mathrm{en}}(Z^{(k)})^{:\hat{p}}, m_{\mathrm{en}}(Z^{(k)})^{\hat{p}:}\right),$$

and $\lambda_1$, $\lambda_2$, and $\delta$ are tuning parameters. In our numerical experiments in Section 5, we employ the kernel-based Maximum Mean Discrepancy (MMD, Gretton et al., 2012) for the invariance loss and the Hilbert-Schmidt independence criterion (HSIC, Gretton et al., 2005) for the independence loss. We compare using different kernels for the MMD and HSIC losses, as well as other simpler, non-kernel based losses in Appendix E.3.

## 4. Good and Bad Use of Learned Representations

Having established that the latent component $W$ serves as a valid instrument for identifying the ACE, a natural question arises: Does the invalid component $V$ hold any statistical value? Specifically, can incorporating $V$ into the estimation procedure improve statistical efficiency (reduce variance) without compromising consistency?

**Efficiency Gains in the Ideal Scenario.** Consider the ideal scenario where the true latent factors $W$ and $V$ are directly observed (see Figure 2). Under Setting 2.1, while $W$ is the instrument, $V$ acts as a proxy for the unobserved confounders $H$. Since $V$ is correlated with the outcome $Y$ (via $H$) but independent of the instrument $W$, it qualifies as a valid adjustment covariate. In this setting, we can employ the Covariate-Adjusted Two-Stage Least Squares (2SLS) estimator, where $V$ is included in the adjustment

set (i.e., partialled out from $D$, $Y$, and $W$). Theoretical frameworks for variance reduction in IV models (Davidson & MacKinnon, 1993; Vansteelandt & Didelez, 2018) and recent graphical criteria for optimal adjustment sets (Henckel et al., 2024) confirm that conditioning on such variables generally reduces the asymptotic variance of the estimator. We formally state this result in the linear setting in Proposition B.9.

**Risks with Learned Representations.** In practice, however, we need to rely on learned representations $\widehat{W}$ and $\widehat{V}$. Using these imperfect proxies for adjustment introduces two specific risks. The first is an *efficiency loss via information leakage*. If the disentanglement is imperfect, $\widehat{V}$ may capture some information belonging to $W$. In this case, partialling out $\widehat{V}$ inadvertently removes valid instrumental variation from $\widehat{W}$, thereby weakening the first stage and inflating the variance of the ACE estimator (potentially worse than using no adjustment). The second is a *bias via collider structures*. If the learned representation fails to satisfy the independence constraint, $\widehat{V}$ may even act as a collider (e.g., if it becomes a function of both $W$ and $H$). Conditioning on such a variable opens a backdoor path, rendering common estimators such as 2SLS biasd. We illustrate these two failure modes explicitly in Examples C.5 and C.6 in Appendix C.

*Remark* 4.1. In this work, we focus on 2SLS as the estimator for concreteness. The consistency of 2SLS does not require a structurally linear instrument-exposure relationship, it only requires relevance in the linear projection sense (see, e.g., Newey & Powell, 2003). More broadly, the learned representations developed here can be paired with other IV estimators (e.g., GMM, non-parametric IV) depending on the target estimand and the assumptions one is willing to impose.

# 5. Experiments

We empirically validate our framework on both semi-synthetic data based on the All of Us biobank (Section 5.1) and fully synthetic data (Section 5.2). Code to reproduce all experiments is available at https://github.com/shimenghuang/RL4IV-code.

## 5.1. Deconfounding Genetic Variants from All of Us

We conduct a semi-synthetic experiment using real genetic variants from the All of Us (AoU) Research Hub to demonstrate the efficacy of our framework in removing population confounding for Mendelian Randomization tasks.

**Genotype data generation.** We extract 652 genetic variants in the GLP1R region from individuals of East Asian and African predicted ancestry, sampling 8000 individuals from each population (see Appendix E.1). To construct a semi-synthetic dataset that preserves realistic linkage disequilibrium (LD) while allowing for controlled confounding, we

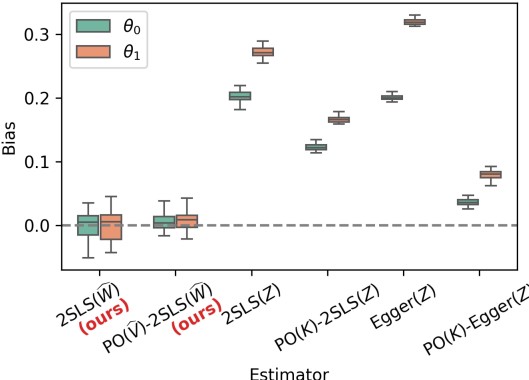

*Figure 3.* Bias of estimated ACE based on different estimators based on semi-synthetic experiments using genetic variants from AoU biobank.

apply Independent Component Analysis (ICA) to the pooled genotype matrix, reducing the data to 10 latent components. We choose the component with the strongest distributional shift across populations to serve as the population-dependent confounder $V$. The remaining components are resampled to ensure invariance, creating the latent component $W$. These representations are then mapped back to the original SNP space, yielding genotypes that exhibit realistic LD structures but possess controlled population stratification profiles.

**Exposures and outcome data.** We generate a two-dimensional exposure $D$ and a univariate outcome $Y$ based on a linear structural causal model (details in Appendix E.1). We introduce an unobserved confounder $H$, generated as a direct copy of $V$, which confounds both the exposure and the outcome. Since the observed instrument $Z$ is generated by $V$, it is naturally confounded by $H$, violating the standard independence assumption. We repeat the sampling of $W$ and all noise variables over 20 random seeds to assess estimator stability.

**Results.** We train our proposed autoencoder with hyper-parameters $\lambda_1 = \lambda_2 = 10$ to recover the invariant representation $\widehat{W}$ and the variant component $\widehat{V}$. We evaluate two estimators (see Definition A.3) based on our learned representations:

1. 2SLS($\widehat{W}$): A standard 2SLS estimator using the learned invariant component as the instrument.
2. PO($\widehat{V}$)-2SLS($\widehat{W}$): The Partialling-Out 2SLS estimator which uses $\widehat{W}$ as the instrument while adjusting for the learned variant component $\widehat{V}$ to improve precision.

We compare these against four baselines: standard 2SLS using the observed $Z$ (2SLS($Z$)), MR Egger (Egger($Z$)), and their population-adjusted counterparts (PO($K$)-2SLS($Z$) and PO($K$)-Egger($Z$)) which partial out the discrete population indicator $K$.

Figure 3 reports the estimation bias for the two causal parameters, $\theta_0$ and $\theta_1$. We see that both of our proposed es-

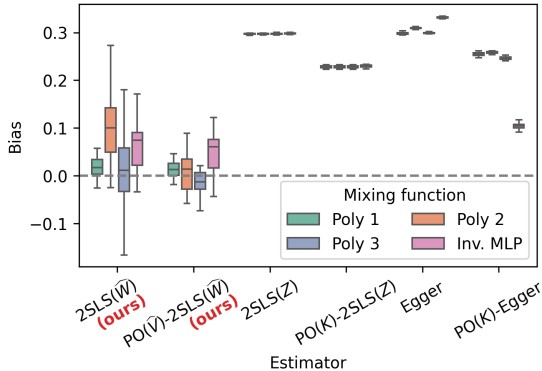

*Figure 4.* Bias comparison under different mixing functions (Polynomials of degree 1-3 and invertible MLP). Our methods remain unbiased while all other methods exhibit large bias.

timators ($2\mathrm{SLS}(\widehat{W})$ and $\mathrm{PO}(\widehat{V})\text{-}2\mathrm{SLS}(\widehat{W})$) achieve a bias centered near zero for both parameters, effectively removing the genetic confounding. The naive estimators $2\mathrm{SLS}(Z)$ and $\mathrm{Egger}(Z)$ exhibit significant bias (up to 0.3). Notably, partialling out the population label ($\mathrm{PO}(K)$) reduces this bias but fails to eliminate it, suggesting that the confounding structure is more complex than a simple mean shift between populations. Moreover, as anticipated by our theoretical analysis in Section 4, our partialling-out estimator $\mathrm{PO}(\widehat{V})\text{-}2\mathrm{SLS}(\widehat{W})$ exhibits visibly tighter variance compared to $2\mathrm{SLS}(\widehat{W})$, confirming that $\widehat{V}$ serves as a useful covariate for variance reduction.

### 5.2. Theory Verification and Ablation Studies

We verify our theoretical findings and evaluate the robustness of our framework using fully simulated data. The data generation process follows the Structural Causal Model (SCM) in (9) with linear structural equations, while the mixing function generating $Z$ varies. We fix the true latent dimensions to $p = q = 2$, with noise variables sampled from multivariate Gaussian distributions with non-diagonal covariance matrices. Hyperparameters $\lambda_1$ (invariance) and $\lambda_2$ (independence) are selected via cross-validation from a grid of 9 values. All experiments are repeated over 20 random seeds. Further implementation details are provided in Appendix E.2.

**Different mixing mechanisms.** We simulate the observed instrument $Z$ using various mixing functions: polynomials of degree $L \in \{1, 2, 3\}$ and an invertible Multi-Layer Perceptron (MLP). The dimension of $Z$ expands with the degree ($d_Z \in \{5, 15, 35\}$ for polynomial degrees 1, 2, and 3, respectively) and is set to $d_Z = 4$ for the invertible MLP. The learned latent dimensions are correctly specified as $\hat{p} = \hat{q} = 2$. The results are summarized in Figure 4. We observe that our proposed estimators ($2\mathrm{SLS}(\widehat{W})$ and $\mathrm{PO}(\widehat{V})\text{-}2\mathrm{SLS}(\widehat{W})$) consistently achieve low or near-zero bias across all mixing mechanisms. In contrast, all other

methods exhibit significant bias. This confirms that standard IV methods and their population-adjusted variants fail when the mixing is non-linear or when the invalidity is driven by latent environmental factors rather than simple discrete indicators.

**Mis-specified latent dimensions.** We investigate the practical challenge where the true dimension of the valid instrument $W$ ($p = 2$) is unknown. We train models specifying $\hat{p} \in \{1, 2, 3, 4\}$ under a polynomial mixing of degree $L = 3$. As shown in Figure 5, with *under-specification* ($\hat{p} < p$), the bias remains small. This aligns with our theorem on partial identification—recovering a linear projection of $W$ is sufficient for valid IV estimation in just- or over-identified linear IV models. However, *over-specification* ($\hat{p} = 4$) leads to increased variance and bias, likely because the excess capacity allows the model to encode noise or leak information from the variant component $V$. This suggests that in practice, a conservative estimate of the latent dimension is preferable if the learned instrument is still strong (e.g., in the linear case, it satisfies the rank condition). In the experiment of Figure 5 we observe that the invariance loss remains small when $\hat{p} \leq p$ and increases when $\hat{p} > p$ (Table 3 in Appendix E.3). This suggests a practical heuristic to estimate $\hat{p}$: starting from a small value at which $\widehat{W}$ achieves invariance during training and increase until invariance no longer holds.

**With and without independence loss** Finally, we validate the theoretical necessity of the independence constraint (Assumption 3.4). We compare models trained with and without the independence loss ($\lambda_2 = 0$). Figure 6 presents the results. In the upper panel (with independence), we see that both estimators are unbiased, confirming successful disentanglement. In the lower panel (without independence), while $2\mathrm{SLS}(\widehat{W})$ remains largely unbiased, the partialling-out estimator $\mathrm{PO}(\widehat{V})\text{-}2\mathrm{SLS}(\widehat{W})$ suffers from significant bias, particularly under complex mixings (Poly 3, MLP). This empirically confirms our theoretical distinction: recovering $W$ (for standard IV) primarily relies on invariance,

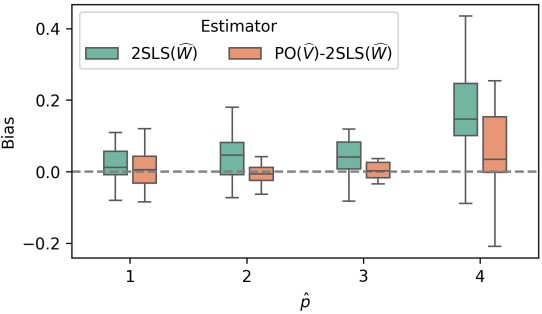

*Figure 5.* Estimation bias given misspecified latent dimensions ($\hat{p}$) when the true dimension is $p = 2$. Our methods remain unbiased for under-specified and moderately over-specified dimensions of $\widehat{W}$.

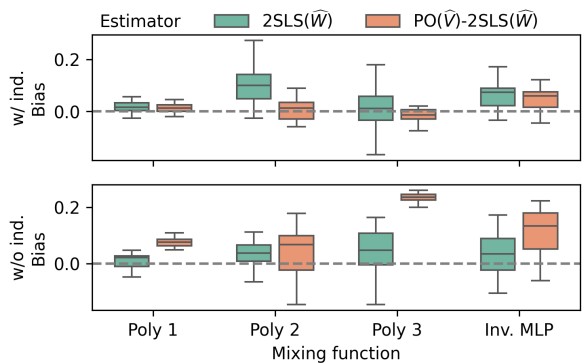

*Figure 6.* Estimation bias with and without including an independence loss. Both 2SLS and PO-2SLS estimators are unbiased when independence loss is included, while PO-2SLS is not unbiased when independence loss is excluded.

but safely using $V$ for variance reduction (via partialling out) strictly requires the independence constraint to prevent collider bias or information leakage.

## 6. Discussion

In this work, we presented a representation learning framework to address an important challenge in Mendelian Randomization: the presence of environmental confounding that renders genetic instruments invalid. By leveraging multi-environment data, we proved that the invariant latent component of a confounded instrument can be recovered up to a transformation sufficient for valid causal inference. We further analyzed the conditions under which the variant component can be safely used to improve estimation efficiency. Moreover, while our framework focuses on multi-environment data (distributional invariance), similar principles apply to multi-view data (paired observations). In such settings, objective functions based on contrastive learning, such as InfoNCE (van den Oord et al., 2018), could replace the MMD loss to exploit sample-level co-occurrence rather than population-level invariance.

**Limitations and Future Work.** Our theoretical guarantees rely on the assumption of sufficient variability (Assumption 3.4), which requires that the distribution of the environmental confounder $V$ shifts in a modular fashion across environments. This assumption is, in general, not testable without further assumptions. A more tractable approach could be to develop falsification procedures: rejecting implications of this assumption may provide evidence against this assumption. Furthermore, in high-dimensional genetic applications, the mapping from latent population structure to observed SNPs is often sparse. Incorporating sparsity constraints into the encoder (e.g., via sparse coding or regularizers as in Liu & Wang (2023)) could improve both interpretability and sample efficiency.

## Acknowledgements

FL is funded partially by the Austrian Science Fund (FWF) 10.55776/COE12. The semi-synthetic experiments used data from the All of Us Research Program. We gratefully acknowledge All of Us participants for their contributions, without whom this research would not have been possible. We also thank the National Institutes of Health's All of Us Research Program for making available the participant data used in this study.

## Impact Statement

This work contributes methodologically to the field of causal inference given observational data. Specifically, it provides theoretical guarantees for identifying valid instrumental variables from environmentally confounded data using multi-environment data. Our framework has potential impact in scientific domains like epidemiology (e.g., Mendelian Randomization) and economics, where it allows researchers to leverage multi-environment data to uncover stable causal mechanisms that were previously obscured. By leveraging representation learning to mitigate potential violations of IV assumptions, this method can lead to more reliable decision-making in healthcare and public policy. However, the validity of our approach hinges on the diversity of the training environments and the assumption of modular distribution shifts. Misapplication to datasets where these assumptions are violated may not lead to a less biased estimate than using the original observed instruments.

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

# Appendix

## A. Additional definitions

We define a Partially Linear Structural Causal Model (PLSCM) for Multi-Environment Mendelian Randomization (MPMR).

**Definition A.1** (PLSCM-MEMR). For any given $k \in [K]$ and for all $i \in [n_k]$

$$
\begin{aligned}
Y_i^{(k)} &:= \left(\theta_0^{(k)}\right)^\top D_i^{(k)} + g^{(k)}(V_i^{(k)}, H_i^{(k)}, \epsilon_i^{(k)}) \\
D_i^{(k)} &:= h^{(k)}(W_i^{(k)}, V_i^{(k)}, H_i^{(k)}, \eta_i^{(k)}) \\
W_i^{(k)} &:= \omega_i^{(k)} \\
V_i^{(k)} &:= l^{(k)}(H_i^{(k)}, \nu_i^{(k)}) \\
Z_i^{(k)} &:= f^{(k)}(W_i^{(k)}, V_i^{(k)}),
\end{aligned}
\tag{9}
$$

where $(\omega_i^{(k)}, \epsilon_i^{(k)}, \eta_i^{(k)}, \nu_i^{(k)}) \sim Q^{(k)}$ and the joint law $Q^{(k)}$ factorizes into the product of the marginal laws of the four variables, i.e., $Q^{(k)} = Q_w^{(k)} \otimes Q_\epsilon^{(k)} \otimes Q_\eta^{(k)} \otimes Q_\nu^{(k)}$. We assume that $Q_w^{(k)} = Q_w$ for all $k \in [K]$, while $Q_\epsilon^{(k)}$, $Q_\eta^{(k)}$, and $Q_\nu^{(k)}$ can be different across $k \in [K]$. $H_i^{(k)} \sim Q_H^{(k)}$ are unobserved variables confounding $V_i^{(k)}$, $D_i^{(k)}$, and $Y_i^{(k)}$. $W_i^{(k)}$ and $V_i^{(k)}$, taking values in $\mathbb{R}^p$ and $\mathbb{R}^q$ respectively, are latent (unobserved) variables which generate the observed variable $Z_i^{(k)}$ via $f^{(k)}$. $f^{(k)}$ is a smooth[4] invertible measurable function, referred to as the *mixing function*, and $g^{(k)}$, $h^{(k)}$ are arbitrary measurable functions. Without loss of generality, we assume that $\mathbb{E}[w_i^{(k)}] = \mathbf{0}$ and $\mathbb{E}[g^{(k)}(V_i^{(k)}, H_i^{(k)}, \epsilon_i^{(k)})] = 0$.

The structural equation for $Z$ is without an additional independent noise, which exists for all other structural equations. This is because any independent noise can be seen as part of $W$, which is not a child of any other variables in the SCM. In essence, in a set of MEMR-PLSCMs, only the distribution of the latent component $W$ is assumed to be unchanged across the environments, while all other variables and parameters may have a changing distribution. Our interest lies in the causal parameters $\theta^{(k)}$ for all $k \in [K]$.

**Definition A.2** (Injective polynomial function). An injective polynomial function of degree $L$, $f : \mathbb{R}^{d_u} \to \mathbb{R}^{d_z}$, is given by

$$
f(u) = A \left[ 1, u^\top, \left(u^{\bar{\otimes} 2}\right)^\top, \cdots, \left(u^{\bar{\otimes} L}\right)^\top \right]^\top,
\tag{10}
$$

where $u^{\bar{\otimes} l}$ denotes the vector of all distinct degree-$l$ monomials in the entries of $u$, and $A \in \mathbb{R}^{d_z \times \binom{L+d_u}{L}}$ has full column rank.

**Definition A.3** (Linear 2SLS estimators). Given i.i.d. obsevations $\{W_i, V_i, D_i, Y_i\}_{i \in [m]}$ where $W_i \in \mathbb{R}^p, V_i \in \mathbb{R}^q, D_i \in \mathbb{R}^d$, and $Y_i \in \mathbb{R}$, denoted as $\mathbf{W} \in \mathbb{R}^{n \times p}$, $\mathbf{V} \in \mathbb{R}^{n \times q}$, $\mathbf{D} \in \mathbb{R}^{n \times d}$, and $\mathbf{Y} \in \mathbb{R}^{n \times 1}$ in matrix form respectively, the linear two-stage least squares (2SLS) estimator is defined as

$$
\hat{\theta}^{\text{2SLS}} := (\mathbf{D}^\top P_W \mathbf{D})^{-1}(\mathbf{D}^\top P_W \mathbf{Y}),
$$

where $P_W := \mathbf{W}(\mathbf{W}^\top \mathbf{W})^{-1}\mathbf{W}^\top$ is the orthogonal projection onto the column space of $\mathbf{W}$. Moreover, let $\widetilde{\mathbf{W}}$, $\widetilde{\mathbf{D}}$, and $\widetilde{\mathbf{Y}}$ denote the residuals of $\mathbf{W}$, $\mathbf{D}$, and $\mathbf{Y}$, respectively, after linearly partialling out $\mathbf{V}$[5]. For instance, $\widetilde{\mathbf{D}} = (I_n - P_V)\mathbf{D}$, where $P_V$ denotes the orthogonal projection onto the column space of $\mathbf{V}$ and $I_n$ is the identity matrix of dimension $n$. We then define the linearly partialled-out 2SLS (PO-2SLS) estimator as

$$
\hat{\theta}^{\text{PO-2SLS}} := (\widetilde{\mathbf{D}}^\top P_{\widetilde{W}} \widetilde{\mathbf{D}})^{-1}(\widetilde{\mathbf{D}}^\top P_{\widetilde{W}} \widetilde{\mathbf{Y}}),
$$

where $P_{\widetilde{W}}$ is the orthogonal projection onto the column space of $\widetilde{\mathbf{W}}$.

## B. Additional theoretical results

**Theorem B.1** (Identification of $W$ given general diffeomorphic mixings and bounded encoder). *Consider Setting 2.1 and assume Assumption 3.10 holds. Let $W$ admit a strictly positive probability density function. Let $\mathcal{A}(\hat{d}_U, \hat{d}_Z)$ be*

---

[4]Unless otherwise specified, in this work, smooth functions refer to $C^1$ functions.

[5]Alternatively, one may partial out $V$ nonlinearly by estimating $\mathbb{E}[T \mid V]$ with a flexible function $\hat{f}$ and setting $\widetilde{\mathbf{D}} = \mathbf{D} - \hat{f}(\mathbf{V})$. In this case, sample splitting is typically required. This approach is known as double machine learning (Chernozhukov et al., 2018).

the set of autoencoders $(m_{en}, m_{de})$ where $m_{en} : \mathcal{Z} \to (a, b)^{\hat{d}_U}$ is a smooth function mapping to an open hypercube, satisfying the reconstruction identity (5) and the invariance constraint (6) for a dimension $\hat{p} \leq \hat{d}_U$. If an autoencoder $(m_{en}, m_{de}) \in \mathcal{A}(\hat{d}_U, \hat{d}_Z)$ maximizes the differential entropy $H(m_{en}(Z)^{:\hat{p}})$, then $m_{en}(Z)^{:\hat{p}}$ depends only on $W$. Moreover,

1) If $\hat{p} = p$, it perfectly identifies $W$ (up to a scalar coordinate-wise diffeomorphism).

2) If $\hat{p} < p$, it partially identifies $W$ (extracts maximal information for that dimension).

*Proof.* Let $h(z) := m_{en}(z)^{:\hat{p}}$ denote the relevant part of the encoder output, and let $\widehat{W} := h(Z)$. The range of $h$ is the bounded open hypercube $\mathcal{C} = (a, b)^{\hat{p}}$.

By Assumption 3.10 (i), there exists a cover $\mathcal{S}$ of $[q]$ where $V$ changes modularly while $W$ and $V^{-S}$ remain invariant. Since the encoder satisfies the invariance constraint (6), we have $h(Z^{(k_1)}) \overset{d}{=} h(Z^{(k_2)})$ for all environments. Applying Assumption 3.10 (ii), any smooth function of $(W, V)$ that is invariant in distribution across these modular shifts must have a zero gradient with respect to $V$. Thus, $h$ depends only on $W$. We can write $\widehat{W} = \psi(W)$ for some smooth function $\psi : \mathbb{R}^p \to \mathcal{C}$.

The autoencoder also maximizes the differential entropy $H(\widehat{W})$ subject to the support constraint $\text{supp}(\widehat{W}) \subseteq (a, b)^{\hat{p}}$. By the result in information theory that the distribution maximizing differential entropy over a bounded region is the Uniform distribution over that region (e.g., Cover, 1999), the maximum entropy is achieved if and only if:

$$\psi(W) \sim \text{Uniform}((a, b)^{\hat{p}}). \tag{11}$$

Assume the class $\mathcal{A}$ is rich enough. By the Darmois-Sklar theorem (Rosenblatt, 1952), there exists a smooth triangular map transforming any strictly positive density on $\mathbb{R}^p$ to the uniform density on $(0, 1)^p$, so a solution exists for $\hat{p} \leq p$.

Let $p_W(w)$ be the density of $W$ and $p_U(u)$ be the uniform density on $\mathcal{C}$. Since $\widehat{W} = \psi(W)$, by the change of variables formula:

$$p_W(w) = p_U(\psi(w)) \cdot |\det J_\psi(w)|,$$

where $J_\psi \in \mathbb{R}^{\hat{p} \times p}$ is the Jacobian of $\psi$.

**Case $\hat{p} = p$ (Perfect Identification):**

Since $\psi(W)$ is Uniform, $p_U(\cdot)$ is a non-zero constant. Thus:

$$|\det J_\psi(w)| \propto p_W(w).$$

Since $p_W(w) > 0$ everywhere (by assumption), the Jacobian determinant is non-zero everywhere. By the Inverse Function Theorem, $\psi$ is a local diffeomorphism. Furthermore, following Zimmermann et al. (2021, Prop. 5), a smooth map from a simply connected domain (like $\mathbb{R}^p$) to a bounded simply connected domain (like $(a, b)^p$) that pushes a strictly positive density to a strictly positive density (Uniform) must be a global diffeomorphism (bijective). Since $\psi$ is invertible, $\sigma(\widehat{W}) = \sigma(W)$, satisfying Definition 3.1 for perfect identification.

**Case $\hat{p} < p$ (Partial Identification):**

In this case, $\psi$ projects $\mathbb{R}^p$ onto a lower-dimensional cube. The entropy maximization ensures that $\widehat{W}$ fills the cube uniformly, implying that the encoder does not collapse information unnecessarily (i.e., it has full row rank $\hat{p}$ almost everywhere). Thus, it captures a subspace of $W$ of dimension $\hat{p}$.

$\square$

*Remark* B.2. If $\hat{p} > p$, the image of $\psi$ is a $p$-dimensional manifold embedded in $\mathbb{R}^{\hat{p}}$. The differential entropy with respect to the $\hat{p}$-dimensional Lebesgue measure is $-\infty$. Thus, the maximization problem is ill-posed or degenerate unless constrained to the manifold, which prevents achieving the Uniform distribution on the full cube. Consequently, we generally assume $\hat{p} \leq p$ for this entropy objective to be meaningful.

**Lemma B.3** (Strict Subadditivity of Concave Functions). *Let $f : [0, \infty) \to \mathbb{R}$ be a concave function satisfying $f(0) = 0$. Then for any finite set of nonnegative numbers $\{x_i\}_{i=1}^n$, it holds that*

$$\sum_{i=1}^n f(x_i) \geq f\left(\sum_{i=1}^n x_i\right).$$

*Moreover, if $f$ is strictly concave over the interval $(0, \sum x_i]$ and there exist at least two indices $j, k$ such that $x_j, x_k > 0$, then the inequality is strict.*

*Proof.* Let $S = \sum_{i=1}^{n} x_i$. If $S = 0$, the inequality holds trivially as $0 \geq 0$. Assume $S > 0$. Let $\lambda_i = \frac{x_i}{S} \in [0, 1]$. Note that $\sum \lambda_i = 1$. Using the concavity of $f$ and the property $f(0) = 0$:

$$
\begin{aligned}
f(x_i) &= f(\lambda_i S + (1 - \lambda_i)0) \\
&\geq \lambda_i f(S) + (1 - \lambda_i)f(0) \quad \text{(Jensen's inequality)} \\
&= \lambda_i f(S).
\end{aligned}
$$

Summing this inequality over $i \in [n]$ yields:

$$
\sum_{i=1}^{n} f(x_i) \geq \sum_{i=1}^{n} \lambda_i f(S) = \left( \sum_{i=1}^{n} \lambda_i \right) f(S) = f(S).
$$

If $x_j, x_k > 0$, then $\lambda_j, \lambda_k \in (0, 1)$. If $f$ is strictly concave, then Jensen's inequality $f(\lambda S + (1 - \lambda)0) \geq \lambda f(S)$ becomes strict. Consequently, the final sum satisfies the strict inequality. $\qquad \square$

**Lemma B.4** (Injectivity and Differential Entropy). *Let $W$ be a random variable in $\mathbb{R}^p$ with an absolutely continuous distribution and finite differential entropy $H(W)$. Let $\mathcal{U}$ be a class of smooth functions $u : \mathbb{R}^p \to \mathbb{R}^p$ with bounded derivatives. Assume $\mathcal{U}$ contains at least one diffeomorphism $u^*$ satisfying a* Jacobian dominance *condition:*

$$
\mathbb{E}_W \left[ \log | \det \nabla u^*(W)| \right] \geq \sup_{u \in \mathcal{U}_{ni}} \mathbb{E}_W \left[ \log | \det \nabla u(W)| \right], \tag{12}
$$

*where $\mathcal{U}_{ni}$ is the subset of functions in $\mathcal{U}$ that are not injective on the support of $W$. Then, for any $u \in \mathcal{U}_{ni}$, it holds that:*

$$
H(u^*(W)) > H(u(W)).
$$

*Proof.* We analyze the differential entropy of the pushforward distribution under both the injective and non-injective maps. Since $u^* \in \mathcal{U}$ is a diffeomorphism, the change of variables formula gives the density of $\widetilde{W}^* := u^*(W)$ as:

$$
p_{\widetilde{W}^*}(y) = p_W(w)| \det \nabla u^*(w)|^{-1}, \quad \text{where } w = (u^*)^{-1}(y).
$$

The entropy is computed directly:

$$
\begin{aligned}
H(u^*(W)) &= - \int p_{\widetilde{W}^*}(y) \log p_{\widetilde{W}^*}(y) dy \\
&= - \int p_W(w) \log \left( \frac{p_W(w)}{| \det \nabla u^*(w)|} \right) dw \quad \text{(by change of variables } y = u^*(w)) \\
&= H(W) + \mathbb{E}_W \left[ \log | \det \nabla u^*(W)| \right].
\end{aligned}
$$

This value is finite by assumption.

Let $u \in \mathcal{U}_{ni}$ be a non-injective map. Let $\widetilde{W} := u(W)$. Since $u$ is smooth, by Sard's theorem, the set of critical values (images of points where $\det \nabla u(w) = 0$) has Lebesgue measure zero. For regular values $y$, the Area Formula gives the density:

$$
p_{\widetilde{W}}(y) = \sum_{w \in u^{-1}(y)} \frac{p_W(w)}{| \det \nabla u(w)|}.
$$

Define the term for a specific pre-image $w$ as $\alpha_w(y) := \frac{p_W(w)}{| \det \nabla u(w)|}$. Then $p_{\widetilde{W}}(y) = \sum_{w \in u^{-1}(y)} \alpha_w(y)$. Consider the function $g(x) = -x \log x$. We evaluate the entropy:

$$
H(\widetilde{W}) = \int_{\text{Im}(u)} g \left( \sum_{w \in u^{-1}(y)} \alpha_w(y) \right) dy.
$$

Since $u$ is not injective on the support of $W$, there exists a set of $y$ with positive measure where the cardinality $|u^{-1}(y)| \geq 2$. On this set, since $p_W$ is strictly positive (by the absolute continuity assumption), the sum contains at least two positive terms. Because $g(x)$ is strictly subadditive for positive arguments (Lemma B.3), we have the strict inequality:

$$
g \left( \sum_{w \in u^{-1}(y)} \alpha_w(y) \right) < \sum_{w \in u^{-1}(y)} g(\alpha_w(y)).
$$

Integrating both sides over the codomain:

$$
\begin{aligned}
H(\widetilde{W}) &< \int_{\mathrm{Im}(u)} \sum_{w \in u^{-1}(y)} g(\alpha_w(y)) dy \\
&= \int_{\mathrm{Im}(u)} \sum_{w \in u^{-1}(y)} -\frac{p_W(w)}{|\det \nabla u(w)|} \log\left(\frac{p_W(w)}{|\det \nabla u(w)|}\right) dy \\
&= \int_{\mathrm{supp}(W)} -p_W(w) \log\left(\frac{p_W(w)}{|\det \nabla u(w)|}\right) dw \quad \text{(by the Area Formula)} \\
&= H(W) + \mathbb{E}_W\left[\log|\det \nabla u(W)|\right].
\end{aligned}
$$

Combining the above with the Jacobian dominance assumption (12):

$$
\begin{aligned}
H(u(W)) &< H(W) + \mathbb{E}_W\left[\log|\det \nabla u(W)|\right] \\
&\leq H(W) + \mathbb{E}_W\left[\log|\det \nabla u^*(W)|\right] \\
&= H(u^*(W)).
\end{aligned}
$$

Thus, $H(u^*(W)) > H(u(W))$. $\qquad \square$

*Remark* B.5. The Jacobian dominance condition (12) is an assumption on the neural network that one uses, it essentially requires that the encoder class $\mathcal{A}$ cannot arbitrarily "inflate" the volume of the latent space to artificially boost entropy. For example, architectures with bounded Lipschitz constants, such as networks trained with Spectral Normalization (Miyato et al., 2018). Since the determinant is the product of singular values, bounding the spectral norm of the weight matrices implies a strict upper bound on the expansion of the volume element (Behrmann et al., 2019), thereby preventing non-injective maps from arbitrarily inflating the entropy.

**Theorem B.6** (Identification of $W$ via Entropy Maximization). *Consider Setting 2.1 and assume Assumption 3.10 holds. Let $\mathcal{A}$ be a class of autoencoders satisfying the reconstruction and invariance constraints with latent dimension $\hat{p} = p$. Assume $\mathcal{A}$ contains at least one encoder $m_{en}^*$ such that $h^*(w) \coloneqq m_{en}^*(f(w,v))^{:p}$ is a diffeomorphism on the support of $W$, and that this encoder satisfies the Jacobian dominance condition (12) relative to any non-injective candidate in $\mathcal{A}$. Then, any autoencoder in $\mathcal{A}$ that maximizes the differential entropy of the valid latent component, $H(m_{en}(Z)^{:p})$, perfectly identifies $W$. That is, the learned representation is a diffeomorphism of the true $W$.*

*Proof.* Let $(m_{\mathrm{en}}, m_{\mathrm{de}}) \in \mathcal{A}$ be an autoencoder maximizing the differential entropy $H(m_{\mathrm{en}}(Z)^{:\hat{p}})$. Let $h(z) \coloneqq m_{\mathrm{en}}(z)^{:\hat{p}}$ denote the valid latent representation. Since $Z = f(W, V)$, we can define the composite function $\psi(w, v) \coloneqq h(f(w, v))$.

First, we establish that the learned representation depends only on $W$. By the invariance constraint (6), the distribution of $\psi(W, V)$ is invariant across all environments. Assumption 3.10 guarantees the existence of a collection of subsets covering $[q]$ where $V$ changes in a modular fashion (while $W$ remains invariant). Applying the Completeness condition of Assumption 3.10 (ii) (as established in the proof of Theorem 3.11), any smooth function of $(W, V)$ that remains distributionally invariant under these modular shifts must have a zero gradient with respect to $V$. Consequently, $\psi(w, v)$ is constant with respect to $v$, and we can write $\psi(w, v) = \phi(w)$ for some smooth function $\phi : \mathbb{R}^p \to \mathbb{R}^{\hat{p}}$. Thus, the encoder output is a deterministic function of $W$ alone: $\widehat{W} = \phi(W)$.

The optimization problem reduces to finding a function $\phi$ in the induced class of functions $\mathcal{U}_\phi = \{m_{\mathrm{en}}(f(\cdot, v))^{:\hat{p}} \mid m_{\mathrm{en}} \in \mathcal{A}\}$ that maximizes $H(\phi(W))$. By assumption, the class $\mathcal{A}$ contains at least one encoder such that the corresponding $\phi^*$ is a diffeomorphism, and this $\phi^*$ satisfies the Jacobian dominance condition (12). We proceed by contradiction. Assume the entropy-maximizing function $\widehat{\phi}$ is not injective almost everywhere on the support of $W$. By Lemma B.4, there exists an injective function $\phi^* \in \mathcal{U}_\phi$ (the diffeomorphism guaranteed by assumption) such that:

$$
H(\phi^*(W)) > H(\widehat{\phi}(W)).
$$

This contradicts that $\widehat{\phi}$ maximizes the entropy. Therefore, the optimal function $\widehat{\phi}$ must be injective almost everywhere.

Since $\widehat{\phi} : \mathbb{R}^p \to \mathbb{R}^p$ is a smooth injective map (and under standard regularity conditions for diffeomorphisms), it is a bi-measurable bijection onto its image. Thus, $\sigma(\widehat{W}) = \sigma(\phi(W)) = \sigma(W)$. This satisfies the definition of perfect identification: $W$ is identified up to an invertible transformation (a diffeomorphism). $\qquad \square$

*Remark* B.7. The Jacobian dominance condition is necessary because differential entropy can be increased in two ways: by expanding the volume of the support (increasing $\mathbb{E}[\log \det J]$) or by preventing the probability mass from overlapping (injectivity). Without a constraint on the volume expansion (e.g., a bounded domain, volume-preserving flows, or a regularizer), a non-injective map could arbitrarily increase entropy by simply scaling up the space. The condition ensures that the maximization objective favors the topological property of injectivity rather than mere geometric expansion.

*Remark* B.8 ((Relationship between Theorems B.1 and B.6)). Theorem B.6 describes the fundamental geometric principle required for identification via entropy maximization: the encoder class must not allow non-injective maps to artificially inflate the latent volume (and thus entropy) more than an injective diffeomorphism could (the Jacobian Dominance condition). Theorem B.1 presents a practical way to enforce this condition by using a specific architecture: a bounded encoder. When the latent space is bounded to a hypercube, the "volume" is fixed. Consequently, the only way to maximize differential entropy is to maximize uniformity (filling the space). Since any non-injective map creates regions of higher density and thus lower entropy compared to an injective map, the bounded constraint ensures that the global maximum of the entropy objective corresponds to an injective map, thereby identifying $W$.

**Proposition B.9** (Efficiency of 2SLS using oracle $V$ and $W$). *Under the SCM in* (9), *the estimator* $\hat{\theta}^{PO\text{-}2SLS}$ *which partials out* $V$ *has a lower asymptotic variance than* $\hat{\theta}^{2SLS}$ *which ignores* $V$, *as long as* $V$ *and* $Y$ *are marginally dependent.*

*Proof.* We first show the asymptotic variance of $\hat{\theta}^{2SLS} = (\mathbf{D}^\top P_W \mathbf{D})^{-1}(\mathbf{D}^\top P_W \mathbf{Y})$, where $P_W = \mathbf{W}(\mathbf{W}^\top \mathbf{W})^{-1}\mathbf{W}^\top$. From (9), we have that $\mathbf{Y} = \mathbf{D}^\top \theta_0 + g(\mathbf{V}, \mathbf{H}, \boldsymbol{\epsilon})$, which gives

$$\hat{\theta}^{2SLS} - \theta_0 = (\mathbf{D}^\top P_W \mathbf{D})^{-1}\mathbf{D}^\top P_W g(\mathbf{V}, \mathbf{H}, \boldsymbol{\epsilon}). \tag{13}$$

Assume the regularity conditions 1), 2), and 4) in Assumption D.1 hold, we have

$$\lim_{n\to\infty} (\frac{1}{n}\mathbf{D}^\top P_W \mathbf{D}) = \lim_{n\to\infty} \left(\frac{1}{n}\mathbf{D}^\top \mathbf{W}\right) \cdot \lim_{n\to\infty} \left(\frac{1}{n}\mathbf{W}^\top \mathbf{W}\right)^{-1} \cdot \lim_{n\to\infty} \left(\frac{1}{n}\mathbf{W}^\top \mathbf{D}\right)$$
$$\xrightarrow{p} \mathbb{E}[DW^\top]\mathbb{E}[WW^\top]^{-1}\mathbb{E}[WD^\top]$$

which exists, is finite, and is positive definite. Then by the continuous mapping theorem, $\lim_{n\to\infty} (\mathbf{D}^\top P_W \mathbf{D})^{-1} \xrightarrow{p} \left(\mathbb{E}[D^\top W]\mathbb{E}[WW^\top]^{-1}\mathbb{E}[WD^\top]\right)^{-1}$. Therefore, other than $\mathbf{W}^\top g(\mathbf{V}, \mathbf{H}, \boldsymbol{\epsilon})$, all terms in (13) have a proper probability limit.

Under the assumption in Setting 2.1 that $\mathbb{E}[g(V, H, \epsilon)] = 0$, we have that

$$\mathbb{E}[Wg(V, H, \epsilon)] = \mathbb{E}[W \mid g(V, H, \epsilon)]\mathbb{E}[g(V, H, \epsilon)] = 0.$$

Given conditions 6)-7) in Assumption D.1, we get by the central limit theorem that

$$\frac{1}{\sqrt{n}}\mathbf{W}^\top g(\mathbf{V}, \mathbf{H}, \boldsymbol{\epsilon}) \xrightarrow{d} \mathcal{N}(\mathbf{0}, \sigma^2 \mathbb{E}[WW^\top]).$$

Therefore, by Slutsky's theorem,

$$\sqrt{n}\left(\hat{\theta}^{2SLS} - \theta_0\right) \xrightarrow{d} \mathcal{N}(\mathbf{0}, \Sigma^{2SLS}),$$

where

$$\Sigma^{2SLS} = \sigma^2 \left(\mathbb{E}[D^\top W]\mathbb{E}[WW^\top]^{-1}\mathbb{E}[WD^\top]\right)^{-1}.$$

With $\hat{\theta}^{PO\text{-}2SLS} = (\widetilde{\mathbf{D}}^\top P_{\widetilde{W}}\widetilde{\mathbf{D}})^{-1}(\widetilde{\mathbf{D}}^\top P_{\widetilde{W}}\widetilde{\mathbf{Y}})$, where $\widetilde{\mathbf{D}} = \mathbf{D} - P_V\mathbf{D}$, $\widetilde{\mathbf{W}} = \mathbf{W} - P_V\mathbf{W}$, and $\widetilde{\mathbf{Y}} = \mathbf{Y} - P_V\mathbf{Y}$, we have

$$\hat{\theta}^{PO\text{-}2SLS} - \theta_0 = (\widetilde{\mathbf{D}}^\top P_{\widetilde{W}}\widetilde{\mathbf{D}})^{-1}\widetilde{\mathbf{D}}^\top P_{\widetilde{W}}\tilde{g}(\mathbf{V}, \mathbf{H}, \boldsymbol{\epsilon})$$

where $\tilde{g}(\mathbf{V}, \mathbf{H}, \boldsymbol{\epsilon}) = (I_n - P_V)g(\mathbf{V}, \mathbf{H}, \boldsymbol{\epsilon})$ whose limiting variance satisfies

$$\tilde{\sigma}^2 := \lim_{n\to\infty} \frac{1}{n}\tilde{g}(\mathbf{V}, \mathbf{H}, \boldsymbol{\epsilon})^\top \tilde{g}(\mathbf{V}, \mathbf{H}, \boldsymbol{\epsilon}) = \lim_{n\to\infty} \frac{1}{n}g(\mathbf{V}, \mathbf{H}, \boldsymbol{\epsilon})^\top (I_n - P_V)g(\mathbf{V}, \mathbf{H}, \boldsymbol{\epsilon})$$
$$= \mathbb{V}(g(V, H, \epsilon)) - \mathbb{E}[g(V, H, \epsilon)V^\top]\mathbb{E}[VV^\top]^{-1}\mathbb{E}[Vg(V, H, \epsilon)]$$
$$\leq \mathbb{V}(g(V, H, \epsilon)).$$

The last inequality follows from that $\mathbb{E}[VV^\top]$ is positive definite, and strict inequality holds if $\mathbb{E}[Vg(V, H, \epsilon)] \neq \mathbf{0}$.

Under the regularity conditions 3) and 5) in Assumption D.1, we have

$$\lim_{n\to\infty}\left(\frac{1}{n}\widetilde{\mathbf{D}}^\top\widetilde{\mathbf{W}}\right) = \lim_{n\to\infty}\left(\frac{1}{n}\mathbf{D}^\top\mathbf{W}\right) - \lim_{n\to\infty}\left(\frac{1}{n}\mathbf{D}^\top P_V\mathbf{W}\right)$$
$$\xrightarrow{p} \mathbb{E}[DW^\top] - \mathbb{E}[DV^\top]\mathbb{E}[VV^\top]^{-1}\mathbb{E}[VW^\top]$$
$$= \mathbb{E}[DW^\top]$$

The last equality follows from $V \perp\!\!\!\perp W$ by assumption in Setting 2.1. Similarly, we get

$$\lim_{n\to\infty}\left(\frac{1}{n}\widetilde{\mathbf{W}}^\top\widetilde{\mathbf{W}}\right)^{-1} \xrightarrow{p} \mathbb{E}[WW^\top]^{-1}$$

Thus, $\lim_{n\to\infty}\left(\frac{1}{n}\widetilde{\mathbf{D}}P_{\widetilde{W}}\widetilde{\mathbf{D}}\right) \xrightarrow{p} \mathbb{E}[DW^\top]\mathbb{E}[WW^\top]^{-1}\mathbb{E}[WD^\top]$ which also exists, is finite, and is positive definite.

$$\sqrt{n}\left(\hat{\theta}^{\text{PO-2SLS}} - \theta_0\right) \xrightarrow{d} \mathcal{N}(\mathbf{0}, \Sigma^{\text{PO-2SLS}}),$$

where

$$\Sigma^{\text{PO-2SLS}} = \tilde{\sigma}^2\left(\mathbb{E}[D^\top W]\mathbb{E}[WW^\top]^{-1}\mathbb{E}[WD^\top]\right)^{-1}.$$

Therefore, we have $\Sigma^{\text{PO-2SLS}} \preccurlyeq \Sigma^{\text{2SLS}}$ and $\Sigma^{\text{PO-2SLS}} \prec \Sigma^{\text{2SLS}}$ if $\mathbb{E}[Vg(V, H, \epsilon)] \neq \mathbf{0}$.

$\square$

# C. Additional Examples

**Example C.1.** *Let $K = 2$. $V^{(1)} \sim \mathcal{N}(\mu^{(1)}, \Sigma^{(1)})$ and $V^{(2)} \sim \mathcal{N}(\mu^{(2)}, \Sigma^{(2)})$ are both 2-dimensional Gaussian random vectors, where $\Sigma^{(1)}$ and $\Sigma^{(2)}$ are positive definite, and $\Sigma^{(2)} = \Sigma^{(1)} + D$ where $D$ is symmetric and positive definite. Then let $u \in \mathbb{R}^2$ be an arbitrary non-zero constant vector, we have $u^\top V^{(1)} \sim \mathcal{N}(u^\top\mu^{(1)}, u^\top\Sigma^{(1)}u)$ and $u^\top V^{(2)} \sim \mathcal{N}(u^\top\mu^{(2)}, u^\top\Sigma^{(2)}u)$. Then, it holds that $u^\top\Sigma^{(1)}u - u^\top\Sigma^{(2)}u = u^\top(\Sigma^{(1)} - \Sigma^{(2)})u = u^\top Du > 0$. Thus, for all $u \in \mathbb{R}^2$ such that $u \neq \mathbf{0}$, $u^\top\Sigma^{(1)}u \neq u^\top\Sigma^{(2)}u$ and $u^\top V^{(1)} \overset{d}{\neq} u^\top V^{(2)}$, while if $u = \mathbf{0}$, $u^\top V^{(1)} = u^\top V^{(2)} = 0$.*

**Example C.2.** *Suppose we have data from $K$ environments. Let $V^k \sim Q_{V^k}$ be a random variable taking values in $\mathbb{R}$, and let $W \sim Q_W$ be a random variable taking values in $\mathbb{R}^2$ where the two coordinates satisfy $W^1 \perp\!\!\!\perp W^2$. The observable is generated as $Z := A\left(V, W^1, W^2\right)^\top$ where $A \in \mathbb{R}^{3\times 3}$ is full rank. Let $\widehat{W} := m_{en}(Z)^1 = \left(W^1\right)$ and $\widehat{V} := m_{en}(Z)^{2:3} = \begin{pmatrix} W^2 \\ V \end{pmatrix}$. Then, let $m_{de}$ be the identity map, we have $m_{de} \circ m_{en}(Z) = Z$, and $\widehat{W}$ satisfies distributional invariance. However, $\widehat{W}$ does not perfectly identify $W$ according to Definition 3.1, although it still partially identifies $W$.*

**Example C.3.** *Consider the setting in Example C.2. Let $\widehat{V} := m_{en}(Z)^1 = V + W^1$ and let $\widehat{W} := m_{en}(Z)^{2:3} = \left(W^1, W^2\right)^\top$. Let $m_{de}(\widehat{V}, \widehat{W}) := \begin{pmatrix} \widehat{V} - \widehat{W}^1 \\ \widehat{W} \end{pmatrix}$, then $\widehat{W}$ satisfies distributional invariance, $m_{de} \circ m_{en}(Z) = Z$ meets the reconstruction identity, and while $\widehat{W}$ perfectly identifes $W$, $\widehat{V}$ does not identify $V$.*

**Example C.4** (Non-identification of $V$ under general diffeomorphic mixing). *Suppose $W, V \sim \mathcal{N}(0, I)$ are standard normal. Let the encoder produce:$\hat{W} = W$, $\hat{V} = R(W)V$, where $R(W)$ is a rotation matrix that changes based on $W$. Here, $\hat{V}$ is marginally $\mathcal{N}(0, I)$ and $\hat{V} \perp\!\!\!\perp \hat{W}$ (because of the spherical symmetry of the Gaussian). However, $\hat{V}$ functionally depends on $W$. Thus, the representations failed to "disentangle" $V$ from $W$ even though all constraints are met.*

**Example C.5** (Reduced efficiency of PO-2SLS using learned $\widehat{W}$ and $\widehat{V}$). *Consider the following SCM whose induced DAG is given in Figure 7:*

$$V := H + \epsilon^V$$
$$W^1 := \epsilon^{W^1}$$
$$W^2 := W^1 + \epsilon^{W^2}$$
$$D := V + W^1 + W^2 + H + \epsilon^D$$
$$Y := \theta D + H + \epsilon^Y,$$

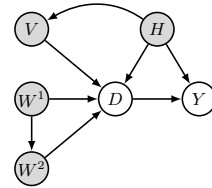

*Figure 7.* DAG induced by the SCM in Example C.5

where $\epsilon^j$ for $j \in \{V, W^1, W^2, D, Y\}$ as well as $H$ are mutually independenet univariate noise variables with finite variances. Suppose $\widehat{V} := m_{en}(Z)^{1:2} = (V, W^1)$ and $\widehat{W} := m_{en}(Z)^3 = W^2$ are the learned representations of $V$ and $W$ respectively. Let $n$ be the number of observations. The asymptotic variance of $\hat{\beta}^{2SLS}$ using $\widehat{W}$ as the instrument can be found as

$$n \cdot \mathbb{V}(\hat{\beta}^{2SLS}) = \mathbb{V}(H + \epsilon^Y) \Big/ \frac{\left(2\mathbb{V}(\epsilon^{W^1}) + \mathbb{V}(\epsilon^{W^2})\right)^2}{\mathbb{V}(\epsilon^{W^1}) + \mathbb{V}(\epsilon^{W^2})}.$$

The asymptotic variance of $\hat{\beta}^{PO\text{-}2SLS}$ using the residual of $\widehat{W}$ after regressing out $\widehat{V}$ is given by

$$n \cdot \mathbb{V}(\hat{\beta}^{PO\text{-}2SLS}) = \frac{\mathbb{V}(H + \epsilon^Y)}{\mathbb{V}(\epsilon^{W^2})}.$$

This gives $\mathbb{V}(\hat{\beta}^{PO\text{-}2SLS}) > \mathbb{V}(\hat{\beta}^{2SLS})$ for any $n$.

**Example C.6.** *Consider the following SCM*

$$\begin{aligned} H &= \epsilon^H \\ V &= H + \epsilon^V \\ W &= \epsilon^W \\ D &= V + W + H + \epsilon^D \\ Y &= \theta D + H + \epsilon^Y, \end{aligned}$$

where $\epsilon^j$ for $j \in \{V, W, D, Y\}$ as well as $H$ are mutually independenet univariate noise variables with finite variances.

Suppose $\widehat{W} = W$ and $\widehat{V} = V + W$. Denote the variances of the noise variables as $\sigma_j^2$ for $j \in \{H, V, W, D, Y\}$ respectively. Let $\widetilde{W}$ denote the residual from regressing $\widehat{W}$ on $\widehat{V}$, similarly, for $\widetilde{D}$ and $\widetilde{Y}$. Then

$$\begin{aligned} \hat{\beta}^{PO-2SLS} \xrightarrow{p} & \frac{Cov(\widetilde{W}, \widetilde{Y})}{Cov(\widetilde{W}, \widetilde{D})} \\ &= \frac{\sigma_H^2 \sigma_W^2 (\theta + 1)}{\sigma_H^2 + \sigma_V^2 + \sigma_W^2} \cdot \frac{\sigma_H^2 + \sigma_V^2 + \sigma_W^2}{\sigma_H^2 \sigma_W^2} \\ &= \theta + 1 \end{aligned}$$

*Therefore the asymptotic bias of $\hat{\beta}^{PO-2SLS}$ is 1.*

## D. Proofs

**Assumption D.1** (Regularity conditions)**.** Under Setting 2.1, for any given $k \in [K]$ (superscript omitted below for clarity), assume the following conditions hold

1) $\lim_{n\to\infty} \left(\frac{1}{n}\mathbf{W}g(\mathbf{V}, \mathbf{H}, \boldsymbol{\epsilon})\right) \xrightarrow{p} \mathbb{E}[Wg(V, H, \epsilon)] = \mathbf{0}$

2) $\lim_{n\to\infty} \left(\frac{1}{n}\mathbf{W}^\top\mathbf{W}\right) \xrightarrow{p} \mathbb{E}[WW^\top]$ exists, is finite, and is positive definite

3) $\lim_{n\to\infty} \left(\frac{1}{n}\mathbf{V}^\top\mathbf{V}\right) \xrightarrow{p} \mathbb{E}[VV^\top]$ exists, is finite, and is positive definite

4) $\lim_{n\to\infty} \left(\frac{1}{n}\mathbf{W}^\top\mathbf{D}\right) \xrightarrow{p} \mathbb{E}[WT^\top]$ exists, is finite, and has full column rank

5) $\lim\limits_{n \to \infty} \left( \frac{1}{n} \mathbf{V}^\top \mathbf{D} \right) \xrightarrow{p} \mathbb{E}[VT^\top]$ exists and is finite

6) $\mathbb{V}[g(V, H, \epsilon)] = \sigma^2$ exists, is finite, and full column rank

7) $\mathbb{V}[W g(V, H, \epsilon)]$ exists and is finite.

**Proposition 2.4** (Bijective transformations of a valid instrument are also valid)**.** *If $Z$ is a valid instrument for $D$ with respect to a response $Y$ (Definition 2.1) and $\kappa$ is a bijective measurable function $\kappa : \mathcal{Z} \to \mathcal{Z}'$, $z \mapsto \kappa(z)$. Then $\kappa(Z)$ is also a valid instrument for $D$ with respect to $Y$.*

*Proof.* By Doob–Dynkin lemma (e.g., Billingsley, 1995), if $\kappa$ is a bijective measurable function, then the $\sigma$-algebra generated by $Z$ and the $\sigma$-algebra generated by $\kappa(Z)$ are exactly the same. That is,

$$\sigma(Z) = \sigma(\kappa(Z)).$$

Since the three conditions in Definition 2.1 are marginal dependence, marginal independence, and conditional independence statements of $Z$ respectivley, they also hold for $\kappa(Z)$ (see Dawid, 1980). $\square$

**Proposition 2.3.** *Given an arbitrary random variable $Z \in \mathcal{Z}$, if there does not exist a function $\ell$ such that $Z = \ell(A, B)$ where $A \perp\!\!\!\perp B$ and $A$ satisfies all conditons in Defintion 2.1, then there is no function $\varphi$ such that $\varphi(Z)$ satisfies all three conditions in Defintion 2.1, and thus the ACE is not identifiable given $Z$, $D$, and $Y$.*

*Proof.* We prove the contrapositive. Assume that there exists a measurable function $\varphi$ such that $\varphi(Z)$ is a valid instrument according to Definiton 2.1. Let $A = \varphi(Z)$. Then by the functional representation lemma (Li & El Gamal, 2018), there exists a random variable $B$ and a funciton $\eta$ such that $A \perp\!\!\!\perp B$ and $Z = \eta(A, B)$. This concludes the proof. $\square$

**Corollary 2.5** (Rank perserving bijective transformations of an identifying IV)**.** *Suppose $\varphi(Z)$ which takes values in $\mathbb{R}^p$ is a valid instrument (Definition 2.1) and satisfies the rank condition (2). If $h$ is an invertible affine transformation, i.e., there exists a constant full rank matrix $B \in \mathbb{R}^{p \times p}$ and a constant vector $c \in \mathbb{R}^p$ satisfying*

$$h(\varphi(Z)) = B\varphi(Z) + c,$$

*then $h(\varphi(Z))$ is also a valid instrument and*

$$\mathbb{E}[h(\varphi(Z))(Y - D^\top \theta)] = 0$$

*identifies the causal effect $\theta_0$.*

*Proof.* The output of an intertible affine transformation maintains the rank condition (2). $\square$

**Lemma 3.2** (Functional characterization of identification)**.** *In Definition 3.1, perfect identification is satisfied if and only if there exists a measurable bijection $\delta : supp(U^S) \to supp(\eta(Z))$ such that $\eta(Z) = \delta(U^S)$ a.s.; partial identification is satisfied if and only if there exists a measurable function $\delta$ such that $\eta(Z) = \delta(U^S)$ a.s., but there exists no measurable function $\omega$ such that $U^S = \omega(\eta(Z))$ a.s..*

*Proof.* This proof relies on the Doob-Dynkin Lemma (e.g., Billingsley, 1995): For any two random variables $X$ and $Y$, $\sigma(X) \subseteq \sigma(Y)$ if and only if there exists a measurable function $f$ such that $X = f(Y)$ a.s..

**Part 1: Perfect Identification**

($\Rightarrow$) Assume $\sigma(\eta(Z)) = \sigma(U^S)$. This implies:

 (1) $\sigma(\eta(Z)) \subseteq \sigma(U^S)$: By the Doob-Dynkin lemma, there exists a measurable function $\delta$ such that $\eta(Z) = \delta(U^S)$ a.s.
 (2) $\sigma(U^S) \subseteq \sigma(\eta(Z))$: By the Doob-Dynkin lemma, there exists a measurable function $\omega$ such that $U^S = \omega(\eta(Z))$ a.s.

Substituting (1) into (2), we get $U^S = \omega(\delta(U^S))$ a.s. This implies that $\omega \circ \delta$ is the identity map on the support of $U^S$. Therefore, $\delta$ must be injective and $\omega$ must be its inverse. Thus, $\delta$ is a bijection between the supports (up to null sets).

($\Leftarrow$) Assume $\eta(Z) = \delta(U^S)$ where $\delta$ is a measurable bijection with measurable inverse. Since $\eta(Z)$ is a function of $U^S$, $\sigma(\eta(Z)) \subseteq \sigma(U^S)$. Since $\delta$ is invertible, we can write $U^S = \delta^{-1}(\eta(Z))$. Thus, $U^S$ is a measurable function of $\eta(Z)$, implying $\sigma(U^S) \subseteq \sigma(\eta(Z))$. Combining these gives $\sigma(\eta(Z)) = \sigma(U^S)$.

**Part 2: Partial Identification**

($\Rightarrow$) Assume $\emptyset \subsetneq \sigma(\eta(Z)) \subsetneq \sigma(U^S)$. The inclusion $\sigma(\eta(Z)) \subseteq \sigma(U^S)$ implies, by Doob-Dynkin, that there exists a measurable $\delta$ such that $\eta(Z) = \delta(U^S)$ a.s. The strict inequality $\sigma(\eta(Z)) \neq \sigma(U^S)$ implies that $\sigma(U^S) \not\subseteq \sigma(\eta(Z))$. By the

converse of Doob-Dynkin, this means $U^S$ cannot be written as a measurable function of $\eta(Z)$. Consequently, $\delta$ cannot be invertible (injective) a.s., since if it were, we could construct the inverse $\omega = \delta^{-1}$ to recover $U^S$, which contradicts the strict inclusion.

($\Leftarrow$) Assume $\eta(Z) = \delta(U^S)$ but $\delta$ is not invertible (specifically, no measurable inverse exists). Since $\eta(Z)$ is a function of $U^S$, we have $\sigma(\eta(Z)) \subseteq \sigma(U^S)$. Since no function $\omega$ exists s.t. $U^S = \omega(\eta(Z))$, we have $\sigma(U^S) \not\subseteq \sigma(\eta(Z))$. Thus, $\sigma(\eta(Z)) \subsetneq \sigma(U^S)$.

$\square$

**Theorem 3.6** (Identification of $W$ under polynomial mixing). *Consider Setting 2.1 and assuming Assumption 3.3 and Assumption 3.4 hold. An autoencoder $(m_{en}, m_{de})$ where $m_{de}$ is also an injective polynomial of degree $L$ (Definition A.2), which satisfies the reconstruction identity (5) and the invariance constraint (6) with $\hat{p}$, satisfies that $m_{en}(Z)^{:\hat{p}}$ perfectly identifies $W$ if $\hat{p} \geq p$. More specifically, $m_{en}(Z)^{:\hat{p}}$ identifies $W$ up to an affine transformation. That is, there exist a constant matrix $A \in \mathbb{R}^{\hat{p} \times p}$ with full column rank and a constant vector $a \in \mathbb{R}^{\hat{p}}$ such that*

$$m_{en}(Z^{(k)})^{:\hat{p}} = AW^{(k)} + a$$

*for all $k \in [K]$. Moreover, $m_{en}(Z)^{:\hat{p}}$ partially identifies $W$ if $\hat{p} < p$ (provided the invariant latent dimensions are non-degenerate). Specifically, there exist a constant matrix $A' \in \mathbb{R}^{\hat{p} \times p}$ and a constant vector $a' \in \mathbb{R}^{\hat{p}}$ such that*

$$m_{en}(Z^{(k)})^{:\hat{p}} = A'W^{(k)} + a'$$

*for all $k \in [K]$.*

The proof shares similarities with Theorem 2 in Ahuja et al. (2024), but with the following key differences. First, under Setting 2.1, the latent variables are partitioned into two sets, $V$ and $W$, which are mutually independent. Interventions induced by the hidden variable $H$ affect only the distribution of $V$, and due to the independence between $V$ and $W$, the distribution of $W$ remains unchanged. Second, the interventions on $V$ are not restricted to single-node interventions but may involve arbitrary multi-node interventions.

*Proof.* By Theorem 1 in Ahuja et al. (2024), there exists an invertible matrix $B \in \mathbb{R}^{d_{en} \times (p+q)}$ and a vector $c \in \mathbb{R}^{d_{en}}$ such that $m_{en}(Z) = BU + c$, where $U = (W^\top, V^\top)^\top$. Partition $B$ into blocks corresponding to $W$ and $V$, $B = \begin{pmatrix} B_W & B_D \\ B_C & B_V \end{pmatrix}$, where $B_W := B_{:\hat{p}}^{:p}$, $B_C := B_{\hat{p}:}^{:p}$, $B_D := B_{:\hat{p}}^{p:}$, and $B_V := B_{\hat{p}:}^{p:}$. Consider the first $\hat{p}$ dimension of $m_{en}(Z)$, we have

$$m_{en}(Z)^{:\hat{p}} = B_W W + B_D V + c^{:\hat{p}}.$$

Since the autoencoder satisfies the invariance constraint, for any $k_1, k_2 \in [K]$, we have

$$B_W W^{(k_1)} + B_D V^{(k_1)} = B_W^{:p} W^{(k_2)} + B_D V^{(k_2)}.$$

With $W \perp\!\!\!\perp V$, we can factorize the characteristic functions of the above as

$$\phi_{B_W W^{(k_1)}}(t) \cdot \phi_{B_D V^{(k_1)}}(t) = \phi_{B_W W^{(k_2)}}(t) \cdot \phi_{B_D V^{(k_2)}}(t).$$

Since $W^{(k_1)} \overset{d}{=} W^{(k_2)}$ and since characteristic functions are continuous and equal to 1 at the origin, $\phi_{B_W W}(t)$ is non-zero in a neighborhood of 0, allowing us to divide it out, we obtain

$$B_D V^{(k_1)} \overset{d}{=} B_D V^{(k_2)}.$$

Let $\mathcal{S}$ be the covering collection from Assumption 3.4. Pick an arbitrary $S \in \mathcal{S}$, by Assumption 3.4, there exists $k_1, k_2 \in [K]$ such that

$$B_D^S V^{S,(k_1)} + B_D^{-S} V^{-S,(k_1)} \overset{d}{=} B_D^S V^{S,(k_2)} + B_D^{-S} V^{-S,(k_2)},$$

where $B_D^S$ denotes the columns of $B_D$ indexed by $S$.

Now, since $V^S \perp\!\!\!\perp V^{-S}$ and $V^{-S,(k_1)} \overset{d}{=} V^{-S,(k_2)}$ by Assumption 3.4 (i), we have

$$\phi_{B_D^S V^{S,(k_1)}}(t) \cdot \phi_{B_D^{-S} V^{-S,(k_1)}}(t) = \phi_{B_D^S V^{S,(k_2)}}(t) \cdot \phi_{B_D^{-S} V^{-S,(k_2)}}(t),$$

similar as above, we have $\phi_{B_D^S V^{S,(k_1)}}(t) = \phi_{B_D^S V^{S,(k_2)}}(t)$. Thus $B_D^S V^{S,(k_1)} \overset{d}{=} B_D^S V^{S,(k_2)}$. Then, by Assumption 3.4 (ii), we have that $B_D^S = \mathbf{0}$.

Since this holds for all $S \in \mathcal{S}$ and $\bigcup_i S_i = [q]$, we have that $B_D = \mathbf{0}$. Therefore,

$$m_{\text{en}}(Z)^{:\hat{p}} = B_W W + c',$$

where $c' = c^{:\hat{p}}$.

Lastly, if $\hat{p} \geq p$, since $B$ is invertible and $B_D = \mathbf{0}$, $B_W$ must have full column rank $p$. Thus, $W$ is identified up to an affine transformation; if $\hat{p} < p$, then $B_W$ has rank at most $\hat{p}$, which means $W$ is identified up to a partial linear projection.

$\square$

**Corollary 3.12** (Identification of $W$ and $V$ under common polynomial mixing). *Assume Setting 2.1 holds and $W$ has a finite second moment with a positive definite covariance matrix. Suppose Assumptions 3.3 and 3.4 hold. If an autoencoder $(m_{en}, m_{de})$ with $m_{de}$ being an injective polynomial satisfying the conditions of Theorem 3.6 (with $\hat{p} \geq p$) additionally satisfies the* independence constraint:

$$m_{en}(Z^{(k)})^{:\hat{p}} \perp\!\!\!\perp m_{en}(Z^{(k)})^{\hat{p}:}, \tag{7}$$

*then it identifies both $W^{(k)}$ and $V^{(k)}$ up to affine transformations. Specifically:*

$$m_{en}(Z^{(k)})^{:\hat{p}} = AW^{(k)} + a, \quad m_{en}(Z^{(k)})^{\hat{p}:} = BV^{(k)} + b,$$

*where $A$ and $B$ are constant matrices with full column rank.*

*Proof.* By Theorem 3.6, the satisfaction of the reconstruction and invariance constraints implies that there exists an invertible matrix $B \in \mathbb{R}^{\hat{d} \times (p+q)}$ and a constant vector $c$ such that $m_{\text{en}}(Z) = BU + c$, where $U := (W^{\top}, V^{\top})^{\top}$. Furthermore, the matrix $B$ has the block structure:

$$B = \begin{pmatrix} B_W & \mathbf{0} \\ B_C & B_V \end{pmatrix},$$

where $B_W = B_{:\hat{p}}^{:p} \in \mathbb{R}^{\hat{p} \times p}$ corresponds to the identified $W$ component, $B_C \in \mathbb{R}^{(\hat{d}-\hat{p}) \times p}$ represents the potential leakage of $W$ into the second part, and $B_V = B_{\hat{p}:}^{p:(p+q)} \in \mathbb{R}^{(\hat{d}-\hat{p}) \times q}$ corresponds to $V$. Since $B$ is invertible, the diagonal blocks $B_W$ and $B_V$ must have full column rank. We have

$$\widehat{W} := m_{\text{en}}(Z)^{:\hat{p}} = B_W W + c_1$$
$$\widehat{V} := m_{\text{en}}(Z)^{\hat{p}:} = B_C W + B_V V + c_2$$

The independence constraint $\widehat{W} \perp\!\!\!\perp \widehat{V}$ implies that their covariance is zero.

$$\begin{aligned} \mathbf{0} &= \text{Cov}(\widehat{W}, \widehat{V}) \\ &= \text{Cov}(B_W W, B_C W + B_V V) \\ &= B_W \text{Cov}(W, W) B_C^{\top} + B_W \text{Cov}(W, V) B_V^{\top}. \end{aligned}$$

By assumption, $W \perp\!\!\!\perp V$, so $\text{Cov}(W, V) = \mathbf{0}$. Let $\Sigma_W := \text{Cov}(W, W)$, which is positive definite by assumption. The equation simplifies to:

$$B_W \Sigma_W B_C^{\top} = \mathbf{0}.$$

We multiply from the left by $B_W^{\top}$:

$$(B_W^{\top} B_W) \Sigma_W B_C^{\top} = \mathbf{0}.$$

Since $B_W$ has full column rank, $B_W^{\top} B_W$ is invertible. Since $\Sigma_W$ is positive definite, it is also invertible. Therefore, the only solution is:

$$B_C^{\top} = \mathbf{0} \implies B_C = \mathbf{0}.$$

Thus, $\widehat{V} = B_V V + c_2$, where $B_V$ has full column rank. This concludes the proof. $\square$

**Theorem 3.11** (Identification of $W$ given general diffeomorphic mixings). *Consider Setting 2.1 and suppose Assumptions 3.9 and 3.10 hold. Assume further that $W^{(k)} \perp\!\!\!\perp V^{(k)}$ for all $k \in [K]$ and $W^{(k)}$ is invariant across environments. Let $\mathcal{A}(\hat{p}, \hat{q})$ denote the class of autoencoders $(m_{\text{en}}, m_{\text{de}})$ whose encoder $m_{\text{en}} : \mathcal{Z} \to \mathbb{R}^{\hat{p}+\hat{q}}$ is a diffeomorphism and whose decoder $m_{\text{de}}$ is a smooth function. If an encoder $m_{\text{en}} \in \mathcal{A}(\hat{p}, \hat{q})$ satisfies the invariance constraint (6) with its first $\hat{p}$ dimensions, then the first $\hat{p}$ components of the latent representation depend only on $W$. Specifically, there exists a smooth function $\psi : \mathbb{R}^p \to \mathbb{R}^{\hat{p}}$ such that:*

$$m_{\text{en}}(Z)^{:\hat{p}} = \psi(W) \quad \text{almost surely.}$$

*If $\hat{p} \geq p$, $W$ is identified up to a diffeomorphism ($\psi$ is an embedding). If $\hat{p} < p$, $W$ is partially identified ($\psi$ is a projection).*

*Proof.* Let $f$ be the ground truth mixing diffeomorphism defined in Assumption 3.9. Define the composite function $h : \mathbb{R}^{p+q} \to \mathbb{R}^{\hat{p}}$ as:
$$h(w, v) \coloneqq m_{\text{en}}(f(w, v))^{:\hat{p}}.$$
Since $m_{\text{en}}$ and $f$ are smooth, $h$ is smooth. Our goal is to show that $h$ does not depend on $v$, i.e., $\frac{\partial h}{\partial v} = \mathbf{0}$.

Let $\mathcal{S}$ be the covering collection from Assumption 3.10 (i). Pick an arbitrary subset $S \in \mathcal{S}$. By Assumption 3.10 (i), there exist environments $k_1, k_2$ such that the distribution of $V^S$ changes ($V^{S,(k_1)} \overset{d}{\neq} V^{S,(k_2)}$), while the complement $V^{-S}$ is invariant and $W$ is invariant. Furthermore, the block $V^S$ is independent of the invariant block $(W, V^{-S})$ in these specific environments. The invariance constraint on the encoder output implies:
$$m_{\text{en}}(Z^{(k_1)})^{:\hat{p}} \overset{d}{=} m_{\text{en}}(Z^{(k_2)})^{:\hat{p}}.$$
Substituting $Z = f(W, V)$, we have:
$$h(W^{(k_1)}, V^{S,(k_1)}, V^{-S,(k_1)}) \overset{d}{=} h(W^{(k_2)}, V^{S,(k_2)}, V^{-S,(k_2)}).$$

The relation above represents a smooth function $h$ whose output distribution remains invariant despite the input block $V^S$ changing distribution (while the rest of the inputs $W, V^{-S}$ remain invariant and independent of $V^S$). This is exactly the precondition for Assumption 3.10 (ii). Therefore, we conclude that $h$ must be independent of the variables in $S$:
$$\frac{\partial h}{\partial v_j}(w, v) = \mathbf{0} \quad \forall j \in S.$$

Since $\bigcup_{S \in \mathcal{S}} S = [q]$, the condition $\frac{\partial h}{\partial v_j} = \mathbf{0}$ holds for all $j \in \{1, \dots, q\}$. Consequently, the gradient of $h$ with respect to the entire vector $v$ is zero everywhere. This implies that $h(w, v)$ is constant with respect to $v$. Thus, there exists a smooth function $\psi : \mathbb{R}^p \to \mathbb{R}^{\hat{p}}$ such that:
$$h(w, v) = \psi(w).$$

Substituting back the definition of $h$:
$$m_{\text{en}}(Z)^{:\hat{p}} = \psi(W).$$

Since $m_{\text{en}}$ and $f$ are diffeomorphisms, their composition $m_{\text{en}} \circ f$ has full rank $p + q$. The Jacobian of the top $\hat{p}$ components is:
$$J = \begin{pmatrix} \frac{\partial h}{\partial w} & \frac{\partial h}{\partial v} \end{pmatrix} = \begin{pmatrix} \frac{\partial \psi}{\partial w} & \mathbf{0} \end{pmatrix}.$$

Case $\hat{p} \geq p$: For the mapping to preserve as much information as possible (as enforced by the reconstruction loss on the full $p + q$ space), $\psi$ must have rank $p$. Thus, $W$ is identified up to an injective smooth transformation (a diffeomorphism onto its image).

Case $\hat{p} < p$: $\psi$ has rank at most $\hat{p}$, representing a dimensionality reduction of $W$ (partial identification). $\qquad\square$

# E. Experiment Details and Additional Experiments

### E.1. Details of the Semi-synthetic Experiment

**Genotype Data Preprocessing.** We extract genetic variants in the GLP1R region from the All of Us (AoU) Biobank. The genomic region of GLP1R was obtained from GeneCards (2025), which is based on the GRCh38 human genome assembly (Genome Reference Consortium, 2013). This results in $4,830$ variants. We then split multiallelic variants into biallelic variants and combine these with single-nucleotide polymorphisms (SNPs), setting allele frequency thresholds of between $0.01$ and $0.99$. Moreover, to ensure the observations are close to i.i.d., we remove related samples according to the relatedness information provided by AoU, and we subset the East Asian population and African population defined by the predicted genetic ancestry of AoU. Then, for each of the two populations, we further filter out variants that have missing data in more than $1\%$ of the samples and variants that deviate significantly from Hardy-Weinberg Equilibrium with p-value $10^{-6}$. We take the common remaining SNPs between the two populations, resulting in $652$ variants.

**Semi-synthetic Data Generation.** As mentioned in Section 5.1, we perform Independent Component Analysis (ICA) on the pooled genotype matrix, after standardizing it. We take the first component, which differs the most from the two populations,

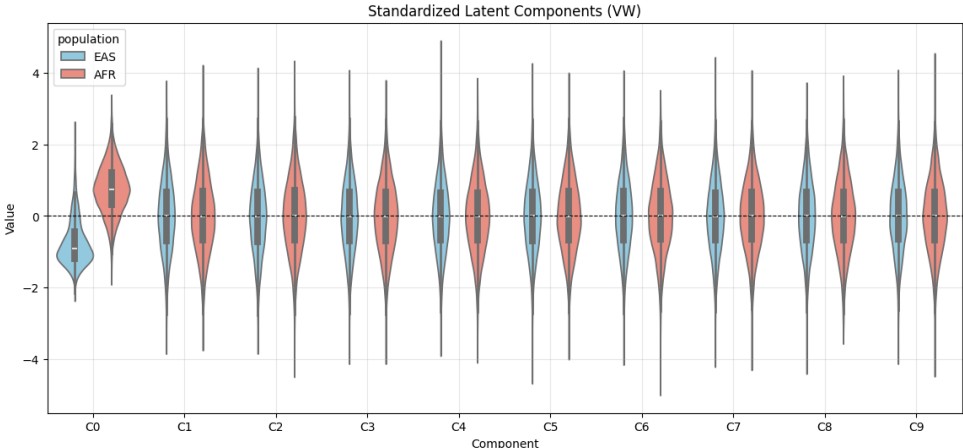

*Figure 8.* Distributions of the latent variables. The first component ($C0$) (obtained from real data) is used as the true $V$ and the rest of the components (simulated) are used as true $W$.

as the true $V$ component, and generate another 9 components from standard Gaussian as the $W$ component to ensure that it has a stable distribution across populations. An example of the resulting distributions of $V$ and $W$ is given in Figure 8.

Based on $V^{(k)}$ and $W^{(k)}$ for $k \in \{1, 2\}$, we simulate other variables based on the following linear SCM:

$$H^{(k)} := V^{(k)}$$
$$D^{(k)} := V^{(k),\top}\beta_1 + W^{(k),\top}\beta_2 + H^{(k),\top}\alpha_1 + \varepsilon_D$$
$$Y^{(k)} := D\theta + H^{(k),\top}\alpha_2 + \varepsilon_Y,$$

where $\varepsilon_D \sim \mathcal{N}(0, I_2)$, $\varepsilon_Y \sim \mathcal{N}(0, 1)$, $\alpha_1 = (0.5, 0.5)^\top$, $\alpha_2 = 0.75$,
$\beta_1 = (-0.5, -0.5, -0.5, -0.5, -0.5, 1.0, 1.0, 1.0, 1.0, 1.0)^\top$, $\beta_2 = (1.0, 1.0, 1.0, 1.0, 1.0, -0.5, -0.5, -0.5, -0.5, -0.5)^\top$,
and $\theta = (1.0, 1.0)^\top$.

### Model Specification.

- **Encoder.** For all experiments in Section 5.2, we employ an MLP encoder that consists of the following layers:
    - Linear($d_z, 200$)
    - Linear($200, \hat{p} + \hat{q}$)
- **Decoder**: we use an MLP decoder that consists of the following layers:
    - Linear($\hat{p} + \hat{q}, 100$)
    - Linear($100, d_z$)

We use Maximum Mean Discrepancy (MMD) for the invariance loss and Hilbert-Schmidt Independence Criterion (HSIC) for the independence loss, both with a polynomial kernel of degree 2, and fix $\lambda_1 = \lambda_2 = 1.0$. We use a batch size of 250 and a total number of 100 epochs. Learning rate is fixed at $10^{-3}$, weight decay is fixed at $10^{-4}$, and gradient clip is set at 1.0.

### E.2. Details of the Ablation Studies

**Notation.** *Let $I_d$ denote the diagonal matrix of dimension $d \times d$ and $\mathbf{1}_m$ denote the vector of ones of dimension $m \times 1$.*

### Data-Generating Process.

We generate data from the following linear SCM:

$$H^{(k)} := \varepsilon_H$$
$$V^{(k)} := \eta^{(k)}H + \varepsilon_V$$
$$W^{(k)} := \varepsilon_W$$
$$D^{(k)} := V^{(k),\top}\beta_1 + W^{(k),\top}\beta_2 + H^{(k),\top}\alpha_1 + \varepsilon_D$$
$$Y^{(k)} := D\theta + H^{(k),\top}\alpha_2 + \varepsilon_Y,$$

where $\varepsilon_H \sim \mathcal{N}(0, I_2)$, $\varepsilon_V \sim \mathcal{N}(0, \Sigma_V)$ and $\varepsilon_W \sim \mathcal{N}(0, \Sigma_W)$ with $\Sigma_V = \Sigma_w = \begin{pmatrix} 1 & 0.5 \\ 0.5 & 1 \end{pmatrix}$, $\varepsilon_D \sim \mathcal{N}(0, I_2)$,

$\varepsilon_Y \sim \mathcal{N}(0,1)$, $\eta^{(1)} = I_2$, $\eta^{(2)} = 2 \cdot I_2$, $\alpha_1 = \alpha_2 = \mathbf{1}_2$, $\beta_1 = \beta_2 = \mathbf{1}_2$, and $\theta = 1.0$.

Then $Z^{(k)}$ is generated from $V^{(k)}$ and $W^{(k)}$, $Z^{(k)} := f(W^{(k)}, V^{(k)})$, based on one of the following *mixing functions*:

- Polynomial of degree 1-3: Injective polynomial function as defined in Definition A.2, implemented by first mapping the input to the polynomial terms of the given degree, then transform them by a single linear layer neural network.
- Invertible MLP: Based on a 2-layer MLP with leaky-ReLU activation function such that the network is approximately invertible. Implementation is taken from the github repository of von Kügelgen et al. (2021).

We fix the seed for all randomness such as weight initialization and the weights in invertible MLP mixing, and we use a sequence of seeds for data generation (20 datasets in each experiment setting). Each dataset contains $10,000$ training samples and $2,000$ validation samples.

**Model Specification.**

- **Encoder.** For all experiments in Section 5.2, we employ an MLP encoder consists of the following layers:
  - Linear($d_z$, 100)
  - LayerNorm
  - ReLU
  - Linear(100, 100)
  - LayerNorm
  - ReLU
  - Linear(100, $\hat{p} + \hat{q}$)
- **Decoder**: For experiments with polynomial mixing, we use a polynomial decoder of the mixing degree, which is implemented by first mapping the input to the polynomial terms of the given degree, then transform them by a single linear layer neural network; for experiments with invertible MLP mixing, we use a MLP decoder consists of the following layers:
  - Linear($\hat{p} + \hat{q}$, 100)
  - ReLU
  - Linear(100, 100)
  - ReLU
  - Linear(100, $d_z$)

**Loss Functions.** The loss function is a weighted sum of the following three terms:

- Reconstruction loss $\mathcal{L}_{\text{rec}}$: Mean Squared Error (MSE).
- Invariance loss $\mathcal{L}_{\text{inv}}$: MMD with polynomial kernel of degree 2.
- Independence loss $\mathcal{L}_{\text{ind}}$: HSIC with polynomial kernel of degree 2.

To prevent the representations from collapsing, we add a small penalty on the log determinant of the covariance matrix of the representations ($m_{\text{en}}(Z)$).

**Hyperparameter in the Loss Function.** For each dataset, we consider the following hyperparameter grid:

- $\lambda_1 \in \{1, 5, 10\}$
- $\lambda_2 \in \{0, 1, 5, 10\}$

which result in 12 combinations. We choose the hyperparameters based on the total loss on the validation set. For experiments without independence loss in **With and without independence loss**, we select the best hyperparameter combination with $\lambda_2 = 0$; otherwise we select from $\lambda_2 \neq 0$.

**Other Hyperparameters.** We usethe Adam optimizer with the following hyperparameters:

- Batch size: 500
- Total epochs: 400
- Learning rate: $10^{-3}$
- Weight decay: $10^{-4}$
- Gradient clip: 1.0

### E.3. Additional Ablation Studies

**More than two populations.** We illustrate the case where we have three populations where the data is generated such that the distribution of $V^1$ differs between populations one and two, and the distribution of $V^2$ differs in populations one and three. In this case, compared to the data-generating process in Appendix E.2, we have $\eta^{(1)} = I_2$, $\eta^{(2} = \begin{pmatrix} 2.0 & 0.0 \\ 0.0 & 1.0 \end{pmatrix}$, and

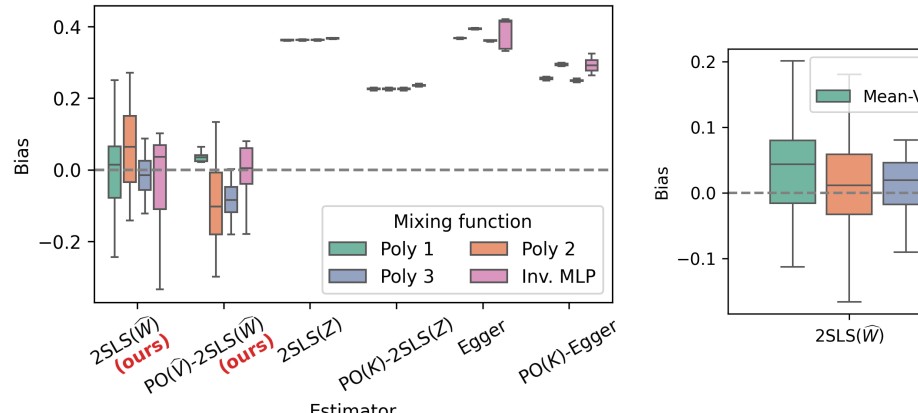

*Figure 9.* Left: Estimation bias in a 3-population experiment given different mixing functions. Right: Estimation bias when using different invariance losses given a polynomial mixing of degree 3.

$\eta^{(3} = \begin{pmatrix} 1.0 & 0.0 \\ 0.0 & 2.0 \end{pmatrix}$. The result is presented in Figure 9 (left), where we compare with other methods as in Section 5.2. We see that the results are similar to Figure 4, and Egger estimators even show a larger variance.

**Different Loss functions** Other than MMD with a polynomial kernel of degree 2, we also conducted experiments with other loss functions for invariance. These include a simple "Mean-Var" loss, which is the sum of L2 norm of the difference in mean and the difference in variance of two samples, and MMD with a polynomial kernel of degree 3. The estimation bias is reported in Figure 9 (right). We see that in this case, the ACE biases are comparable across different invariance losses, for the PO-2SLS estimator, the simple Mean-Var loss even results in the smallest variance.

**Independence loss and the joint effect of both losses.** The above experiments fix the independence loss to HSIC with a degree-2 polynomial kernel. We additionally vary the independence loss across three options: a simple orthogonality penalty (Orth), HSIC with a polynomial kernel of degree 2 (HSIC-Poly 2), and HSIC with a polynomial kernel of degree 3 (HSIC-Poly 3). Combined with the three invariance losses (Mean-Var, MMD-Poly 2, MMD-Poly 3), Tables 1 and 2 report the mean bias (standard deviation across seeds) for $2SLS(\widehat{W})$ and $PO(\widehat{V})$-$2SLS(\widehat{W})$ respectively, across all $3 \times 3$ combinations. The data are generated under a polynomial mixing function of degree 3. HSIC paired with MMD (with either polynomial kernel) yields the lowest bias, while with the two simple penalties, Orth and Mean–Var, the bias is the largest.

*Table 1.* Mean bias (standard deviation across seeds) for $2SLS(\widehat{W})$ under different choices of independence loss (rows) and invariance loss (columns). Data are generated under a polynomial mixing function of degree 3.

| Ind. \ Inv. | Mean–Var | MMD-Poly 2 | MMD-Poly 3 |
|---|---|---|---|
| Orth | 0.242 (0.011) | 0.106 (0.090) | 0.084 (0.025) |
| HSIC-Poly 2 | 0.033 (0.078) | 0.012 (0.094) | 0.013 (0.049) |
| HSIC-Poly 3 | 0.063 (0.184) | 0.038 (0.096) | 0.068 (0.084) |

*Table 2.* Mean bias (standard deviation across seeds) for $PO(\widehat{V})$-$2SLS(\widehat{W})$ under different choices of independence loss (rows) and invariance loss (columns). Data are generated under a polynomial mixing function of degree 3.

| Ind. \ Inv. | Mean–Var | MMD-Poly 2 | MMD-Poly 3 |
|---|---|---|---|
| Orth | 0.233 (0.011) | 0.135 (0.044) | 0.085 (0.023) |
| HSIC-Poly 2 | −0.004 (0.027) | −0.016 (0.026) | −0.003 (0.025) |
| HSIC-Poly 3 | −0.016 (0.035) | −0.009 (0.042) | −0.005 (0.024) |

**Invariance loss across estimated latent dimensions.** For the experiment of Figure 5 (true latent dimension $p = 2$), we additionally record the final invariance loss (MMD between populations on $\widehat{W}$) on validation data. Table 3 reports the mean, minimum, and maximum across seeds for each $\hat{p}$. The invariance loss remains small when $\hat{p} \le p$ and increases significantly when $\hat{p} > p$.

*Table 3.* Final invariance loss (MMD between populations on $\widehat{W}$) on validation data for the experiment of Figure 5 (true dimension $p = 2$).

| $\hat{p}$ | Mean MMD | Min | Max |
|---|---|---|---|
| 1 | 0.0024 | 0.0001 | 0.0058 |
| 2 | 0.0336 | 0.0018 | 0.0997 |
| 3 | 0.1508 | 0.0254 | 0.4258 |
| 4 | 0.9109 | 0.2031 | 2.3672 |

**Effect of a relatedness constraint.** Prior representation-based IV methods such as Cheng et al. (2024a;b) include in their training objective a relatedness term that ties the learned representation to the exposure $D$. Under our framework, identification of the invariant instrument does not require such a constraint: invariance across environments already pins down $\widehat{W}$ as the invariant component of $Z$. In fact, encouraging $\widehat{W}$ to be related to $D$ could harm identification. As $D$ is influenced by the invalid component $V$ through the unobserved confounder $H$, encouraging relatedness may cause $V$ leaking into $\widehat{W}$ and break invariance. To examine this empirically, we add a relatedness term $\lambda_3 \mathcal{L}_{\text{rel}}$ to the loss in Section 3.3, where $\mathcal{L}_{\text{rel}}$ penalizes the lack of association between $\widehat{W}$ and $D$, and vary $\lambda_3 \in \{0, 10^{-3}, 10^{-2}, 10^{-1}\}$. We fix the invariance loss to MMD-Poly 2 and the independence loss to HSIC-Poly 2, and generate data under a polynomial mixing of degree 3. Figure 10 reports the mean bias (with standard deviation across seeds) for 2SLS($\widehat{W}$) and PO($\widehat{V}$)-2SLS($\widehat{W}$). For small $\lambda_3$ (up to $10^{-2}$), the bias remains close to the $\lambda_3 = 0$ baseline but with visibly no gain in efficiency; at $\lambda_3 = 10^{-1}$ both estimators degrade sharply. This is consistent with the intuition that strong relatedness pressure breaks the invariance-based identification of $\widehat{W}$.

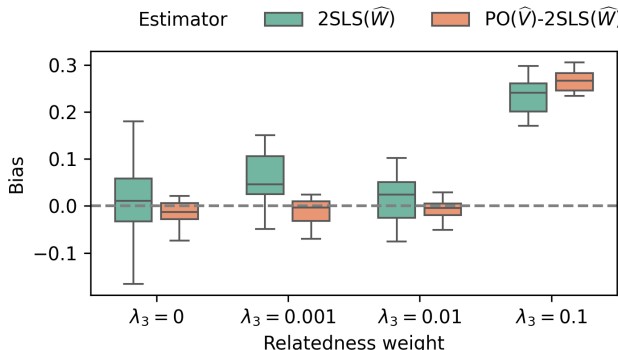

*Figure 10.* Estimation bias (mean with standard deviation across seeds) when an additional relatedness term $\lambda_3 \mathcal{L}_{\text{rel}}$ is added to the training objective. Data are generated under a polynomial mixing of degree 3; the invariance and independence losses are fixed to MMD-Poly 2 and HSIC-Poly 2.

