# OpenReview forum: "Addressing Instrument-Outcome Confounding in Mendelian Randomization through Representation Learning"
_ICML.cc/2026/Conference — ICML 2026 regular_

### Official Review · Reviewer_z2H7 · 2026-02-24

**Soundness:** 3
**Presentation:** 3
**Significance:** 3
**Originality:** 2
**Overall Recommendation:** 4
**Confidence:** 3

**Summary:**

This paper introduced instrumental variable methods to Mendelian randomization research, enabling the decomposition of genetic variants $Z=f(W,V)$ into invariant component $W$ and population dependent component $V$ by training an autoencoder. The paper also included theoretical guarantees under two different settings.

**Compliance With Llm Reviewing Policy:**

Affirmed.

**Final Justification:**

Rebuttal solved my main concerns and suggested some of my questions to be a direction of future work. I will keep my score for this reason.

**Key Questions For Authors:**

In the papar, normalizing flow was mentioned right before Thm 3.11. The paper suggests that it can help identify $W$, but it seems that it's not included in the experiments.

The paper recommends conservative estimation of $p$, can we develop data driven methods to estimate the dimension?

**Limitations:**

yes

**Strengths And Weaknesses:**

Strengths:

The paper focused on a common violation of MR assumptions and proposed a theoretical framework for the problem, the experiments included necessary ablation and demonstrated great performance.

Weaknesses:

The framework follows previous work https://proceedings.mlr.press/v238/ahuja24a/ahuja24a.pdf with similar identification results.

Additional setting with $C^1$ space extends the function space of previous work, but I don't see how this improves estimation, the invertible MLP have larger bias in Figure 4 compared with poly methods.

During All of Us dataset construction, the $V$ component is choosen to be the largerst shift component, and leaving others to be $W$,  this may underestimate the dimensionality and reduce the difficulty of task since the confounding is now a 1D space.

---

> ### Author Rebuttal · Authors · 2026-03-30
>
> We thank the reviewer for their time and constructive feedback. We address the concerns below.
>
> ## Response to weakness:
>
> ### W1: Similarity to Ahuja et al (2024).
>
> **We did make an explicit comparison with Ahuja et al (2024) in Remark 3.5**. Specifically, we differ in that Ahuja et al. (2024) rely on a restrictive assumption of an additive noise SCM over the latent variables and that interventions act solely on the exogenous noise terms. Furthermore, our Assumption 3.4 allows for arbitrary changes in the joint distribution (including the dependence structure) of the block $V^S$, unlike the SCM framework, where distribution shifts are constrained to propagate from independent noise terms.
>
> ### W2: Invertible MLP mixing has a larger bias in Figure 4 compared to polynomial mixings
>
> **The extended function class of the mixing function does not imply that ACE estimation will be improved over the polynomial mixing functions**, as the mixing function is related to how the observed $Z$ is generated. When the observed $Z$ is generated from a non-polynomial mixing function, in this case an invertible MLP, learning representations of the latent variables is likely **more difficult** given finite samples, which explains the larger bias in Figure 4.
>
> ### W3: Semi-synthetic experiment clarification
>
> We agree that the AoU semi-synthetic dataset was created using the largest shifted component as $V$, which may seem restrictive. **This is due to the observed distributions of the components obtained from ICA based on two real populations’ data** (see "Genotype data generation" in Section 5.1), **which do not differ much except for the first component**, and thus, if we use them as the true $V$, they may violate the sufficient variability assumption. That said, we picked another component from ICA that appears to be slightly different in variance across the two populations, and performed another experiment similar to the one shown in Figure 3, with the two-dimensional $V$, for which we provide results in **Table R5** below. We see that the mean bias of our methods (2SLS($\widehat{W}$) and PO($\widehat{V}$)-2SLS($\widehat{W}$)) is the lowest compared to other methods, although compared to the results in Figure 3, the variances are larger. This is due to the fact that the $V^2$ has a smaller difference in distribution, which makes this experiment a more difficult case given finite data.
>
> **Table R5**. Mean bias (std of bias) across seeds for an AoU semi-synthetic experiment with $2$-dimenional $V$.
>
> | Method | $\theta_0$ | $\theta_1$ |
> |---|---|---|
> | Egger($Z$) | 0.2468 (0.0087) | 0.4117 (0.0056) |
> | PO($K$)-Egger($Z$) | 0.1612 (0.0081) | 0.2831 (0.0059) |
> | 2SLS($Z$) | 0.2608 (0.0100) | 0.3753 (0.0077) |
> | PO($K$)-2SLS($Z$) | 0.2281 (0.0058) | 0.3298 (0.0049) |
> | 2SLS($\widehat{W}$) | 0.0747 (0.0957) | 0.1049 (0.1436) |
> | PO($\widehat{V}$)-2SLS($\widehat{W}$) | 0.0681 (0.1072) | 0.1051 (0.1503) |
>
> ## Response to questions:
>
> ### Q1: Other architectures
>
> We thank the reviewer for pointing out the mention of Normalizing Flows (NFs) near Theorem 3.11. **We suggested NFs as a possible architecture due to the recent development of NFs in representation learning**. For example, Li et al (2020) demonstrated the effectiveness of this approach by leveraging the invertibility and exact marginal likelihoods of flows to recover true latent representations. **We chose to focus our experiments on a simpler AE architecture to validate our theory without the computational overhead of flow-based Jacobian calculations**, but we view the integration of these invertible architectures as a valuable direction for future work.
>
> References:
>
> Li, S., Hooi, B., & Lee, G. H. (2020). Identifying through flows for recovering latent representations. Proceedings of the 8th International Conference on Learning Representations (ICLR 2020).
>
> ### Q2: Estimation of latent dimensions
>
> **One possible way to choose $p$ would be starting from a small dimension where $\widehat{W}$ can indeed achieve invariance after training, and increasing it until the learned representation can no longer be invariant across environments**. Note that this only requires the observed data and the learned representations. Below, we provide the **final invariance loss (MMD) in validation data** of the experiment shown in Figure 5 in **Table R3b**. We see that when $\hat{p}$ is larger than the true dimension $p=2$, the invariance loss of the learned representations increases dramatically. This may be an indication that the estimated $\hat{p}$ is too large. We agree that developing a data-driven way to choose $\hat{p}$ is an important direction to further investigate in future work.
>
> **Table R3b**. Final invariance loss (MMD between populations on $\widehat{W}$) in validation data.
>
> | $\hat{p}$ | Mean MMD | Min | Max |
> |---|---|---|---|
> | 1 | 0.0024 | 0.0001 | 0.0058 |
> | 2 | 0.0336 | 0.0018 | 0.0997 |
> | 3 | 0.1508 | 0.0254 | 0.4258 |
> | 4 | 0.9109 | 0.2031 | 2.3672 |

---

> > ### Author Rebuttal · Reviewer_z2H7 · 2026-04-01
> >
> > Thank you for the rebuttal. I tend to keep my score.

---

> > > ### Author Response · Authors · 2026-04-02
> > >
> > > We thank the reviewer for their positive response and for acknowledging that our clarifications have resolved their concerns.

---

### Official Review · Reviewer_KDym · 2026-03-06

**Soundness:** 4
**Presentation:** 4
**Significance:** 4
**Originality:** 4
**Overall Recommendation:** 6
**Confidence:** 3

**Summary:**

This paper proposes a novel methodology to extract valid instruments from invalid instruments using identifiable representation learning and multi environment data. They provide strong theoretical insights into when it is possible to extract a valid instrument from a high dimensional invalid instrument using representation learning and by using results from identifiable representation learning demonstrate when multi environment data can be used to identify the valid instrument. They then provide experiments on semi-synthetic versions of biobank data, demonstrating how their methodology can be applied to remove environmental confounding and extract valid instruments from genetic variants.

**Compliance With Llm Reviewing Policy:**

Affirmed.

**Final Justification:**

I began the reviewing process with a strong assessment of this paper which I maintain. I had minimal concerns which the reviewers addressed.

**Key Questions For Authors:**

Given the paper on the whole is very strong I have few questions for the authors. The only one I will add is that in the limitations section the authors say: "Developing tests to empirically validate the sufficient variability assumption on finite samples is an important direction for future research". Whilst I appreciate the authors mentioning this limitation, I am skeptical that such a test would be possible without relying on some other untestable assumption as we are testing a condition on unobserved variables. Whilst I appreciate it would be nice if it were testable, I would ask the authors if they think it to be realistic that a test for either assumption 3.4 or 3.10 from finite samples of observational variables is realistic?

**Limitations:**

The authors provide a section which clearly addresses the limitations as well as discussing some of the practical limitations with examples.

**Strengths And Weaknesses:**

Whilst not exactly my area of expertise, this appears to be a very strong work with a novel application of identifiable representation learning to mendelian randomisation with multi environment data. The main weaknesses I see of the paper are those which apply more broadly to the fields it draws from (identifiable representation learning) so it seems unfair to hold them against the paper. However I still note them below to give the authors an opportunity to comment on them.

Strengths:
- The theoretical results in this paper are very strong and cover significant ground on this problem. For example proposition 2.3 shows the necessity of their assumed data generating process in order to extract a valid instrument which nicely grounds the assumptions. Section 3 provides a comprehensive selection of results for the identification of the instrument with multi environment data.
- The method follows intuitively from the theory and is well validated with a good semi-synthetic experiment on biobank data.
- The paper is very well presented and easy to follow.

Weaknesses:
- The identifiability relies on a number of assumptions (inevitability, sufficient variability) which are unlikely to hold in practice and may also be impossible to verify. Again as said above I don't hold it against this paper as identifiable representation learning as a whole shares these problems and the authors do note it as an issue, but I think it is worth noting.

---

> ### Author Rebuttal · Authors · 2026-03-30
>
> We sincerely appreciate the positive feedback from the reviewer, and we address the questions below.
>
> ## Response to weakness:
>
> ### W1: Identification assumptions
>
> Indeed, the assumptions we made are common in representation learning. As also noted by another reviewer, the invertibility of the mixing function may not be fully necessary: it may be sufficient if we are able to recover a part of $W$ that is strong enough for ATE estimation. We plan to add a short discussion on this in the paper.
>
> ## Response to questions:
>
> ### Q1: Is it possible to test Assumptions 3.4 or 3.10 from finite samples?
>
> We thank the reviewer for this insightful question. Indeed, we agree that directly testing these assumptions does not seem to be possible without introducing other untestable assumptions. One possible idea in this direction is to investigate ways to falsify this assumption. For example, sufficient viability implies that (unless being cancelled out by the mixing function) the distributions of the observed variable $Z$ should be different. Then, if we reject that the $Z$ has a different distribution, we falsify the sufficient viability assumption. This is, of course, perhaps too weak a test to be useful, but it may at least offer some insight into how different the data is from different environments.

---

> > ### Author Rebuttal · Reviewer_KDym · 2026-04-02
> >
> > I appreciate the authors response to my question. I would still encourage them to relax some of the claims on finite sample tests but this is a minor point that doesn't affect the quality of the paper.

---

> > > ### Author Response · Authors · 2026-04-02
> > >
> > > We appreciate the reviewer's response, and we will make sure to adjust this claim in the limitation section. We thank the reviewer again for their positive and valuable feedback.

---

### Official Review · Reviewer_zeKg · 2026-03-09

**Soundness:** 3
**Presentation:** 2
**Significance:** 3
**Originality:** 3
**Overall Recommendation:** 5
**Confidence:** 4

**Summary:**

The paper presents theory for the identification of latent variables constructed from various mixing functions with applications to Mendelian Randomization. In their setting, instrumental variables are often confounded because they arise from a latent true instrument. The paper presents a few settings where the identification of the latent instrument is possible through representation learning and demonstrate that learning the latent variables improves performance over using the original confounded instrument.

**Compliance With Llm Reviewing Policy:**

Affirmed.

**Final Justification:**

The authors presents theory for the identification of latent variables that can serve as instruments in the Mendelian Randomization setting by removing confounded components of observed variables. Originally, I was concerned by the connection between the theory and experiments and the specificity of assumptions. The authors clarify well in rebuttal which DGPs correspond to which results and the connection of their experiments to their specific loss terms (ie Cor 3.12). The authors also help clarify their work in the context of related work which makes clear their contribution. These address my main concerns, and I now believe this paper is a sound strong original contribution that will lead to further development in this field.

**Key Questions For Authors:**

*Question 1:* Do the synthetic experiments follow from the theoretical results?

*Question 2:* How does the training loss force the latent V to be the confounded representation rather than the instrument?

*Question 3:* How do the architecture and problem setting relate to other mendelian randomization work specifically?

**Limitations:**

Yes

**Strengths And Weaknesses:**

**Strengths**

- The problem setting is thoroughly detailed. The setting has the potential for high impact. (presentation, originality)

- Theoretical results are supported with experiments. Experiments are supported with clear figures showing the superiority of learning latent variables. (presentation)

**Weaknesses**

- My main concern is that it is hard to understand the impact of the theoretical results as written. It would first be helpful to make the use case in Mendelian Randomization more explicit. What are explicit examples of what the variables would be in Example 2.1? What is the environment or domain in this setting? Do these assumptions on the mixing functions make intuitive sense for this setting? To understand the impact of the theory, which makes very structured assumptions, it would help to describe why the assumptions that are made about the latent variables are reasonable in the Mendelian Randomization setting. (presentation, significance)

- Experiments could more directly match the theory. For instance, it would be helpful to highlight data generation processes that match the assumptions and compare the performance of these to other mixing functions. Results in Figure 6 do not seem too different between 2SLS with and without the independence loss. How does this fit with the theory? In addition, how do the architecture and problem setting relate to other mendelian randomization work? Or is this learning setting entirely novel? (soundness, originality)

- It is also not clear how the architecture and loss force learning of the desired latent variables. How does the training loss force the latent V to be the confounded representation rather than the instrument and make latent Z unconfounded? (soundness, presentation)

---

> ### Author Rebuttal · Authors · 2026-03-30
>
> We thank the reviewer for their valuable feedback and recognizing that the setting has potential for high impact.
>
> ## Response to weakness
>
> ### W1: Understanding of assumptions
>
> **Our main motivation is given in Section 1** that observed genetic variants can be viewed as mixtures of invariant biological signals and environment-specific factors (Wang et al., 2020; 2023). **Example 2.1 provides a conceptual basis**: if we recover latent variables $U$ from $Z$, we can isolate valid instruments and remove confounding in the instrument. We defer introducing environments to first highlight this idea before discussing the mechanism for achieving it.
>
> **We focus on a multi-environment data approach as it is natural in MR** (i.e., availability of multi-environment data) and avoids limitations of alternatives: interventional approaches (e.g., Squires et al., 2023) are often infeasible, and ICA (Hyvärinen and Oja, 2000) relies on restrictive assumptions that may not hold in biological settings. We will also clarify the use of “environment” vs. “population”, which we use interchangeably.
>
> References:
>
> Squires, C., Magliacane, S., Kristensen, K., Nielsen, J. S., & Uhler, C. (2023). Causal structure learning from joint interventional and observational data. Journal of Machine Learning Research, 24(178), 1–45.
>
> Hyvärinen, A., & Oja, E. (2000). Independent component analysis: algorithms and applications. Neural networks, 13(4-5), 411-430.
>
> ### W2.1: Experiments' relation to theory
>
> **The experiments are designed to validate our theoretical results in the following way**. They demonstrate how good the ACE estimation is when using $\widehat{W}$ as an instrument (and possibly partialing out $\widehat{V}$): if $\widehat{W}$ does identify $W$ from $X$, ACE bias should be low, as $W$ is a valid instrument (see Proposition 2.4 and Corollary 2.5).
>
> In Section 5.2, “**Different mixing mechanisms**” corresponding to Theorems 3.6, 3.11, and Corollary 3.12, which compares polynomial mixing of different degrees as well as an MLP mixing. “**Mis-specified latent dimensions**” corresponds to Theorem 1.6. With $p=2$, we observe low bias for $\hat{p} = 1,2,3$, but a large bias for $\hat{p} = 4$. As we discussed, this is likely due to the excess capacity allowing the model to leak information of $V$ into $\widehat{W}$. Lastly, “**With and without independence loss**” corresponds to Theorem 1.6 and Corollary 3.12. Since identifying $W$ does not require the independence loss (Theorem 1.6), 2SLS remains low-bias with or without enforcing independence, while PO-2SLS requires $V$ to also be identified, leading to higher bias when independence is not enforced (Figure 6).
>
> ### W2.2: How the proposed method relates to other MR works
>
> As discussed in Section 1, **the problem setting here is to tackle confounding between instruments and outcome using representation learning**, which has not been considered in the MR literature and remains a challenge. Moreover, **although different architectures could be employed for this problem, the key resides in how the model is trained, i.e., the loss**. In Cheng et al (2024), for example, their loss enforces relatedness between the learned representation and the exposure. Note that, as discussed in Section 1.1, existing works do not leverage multi-environment data, nor provide guarantees on the representations learned. Under our framework, identification of the invariant instrument does not require enforcing relatedness, and introducing it can actually harm identification. We will add a result on this in the appendix. Lastly, there exists a recent MR work which explored representation learning (Reddy et al, 2025), but not for the instrument.
>
> Reference:
>
> Reddy, S. G., Cao, F., Xia, R., Loong, S., Chen, E., Steffner, K., ... & Gomes, B. (2025). Deep learning representations and proteome-wide Mendelian randomization identify causal mediators of myocardial fibrosis.
>
> ### W3: Intuition of the loss
>
> **The architecture and the loss function as described in Section 3.3 are chosen to align with our theory in Sections 3.1 and 3.2**. Specifically, Theorems 3.6 and 3.11 state that an autoencoder satisfying the reconstruction identity and the invariance constraint identifies $W$, and Corollary 3.12 states that with an additional independence constraint, $V$ can be identified as well. This is exactly how the loss function (8) was designed, to encourage reconstruction, enforce invariance of $\widehat{W}$ across environments, and enforce independence of $\widehat{W}$ and $\widehat{V}$ in each environment. $\widehat{W}$ and $\widehat{V}$ are identified respectively with this loss, specifically because of the asymmetry between $W$ and $V$, that $\widehat{W}$ is forced to be the invariance part, while $\widehat{V}$ is not.
>
> ## Response to questions:
>
> ### Q1: Theory vs experiments
>
> Please see W2.1.
>
> ### Q2: Loss function
>
> Please see W3.
>
> ### Q3: How do the architecture and setting relate to other MR works
>
> Please see W2.2.

---

> > ### Author Rebuttal · Reviewer_zeKg · 2026-04-01
> >
> > I thank the authors for the clarifications. I will increase my score since my concerns are addressed.
> >
> > Remark: the authors say Thm 1.6 to reference 3.6.

---

> > > ### Author Response · Authors · 2026-04-02
> > >
> > > We deeply appreciate the reviewer's response and are glad we were able to address the concerns.

---

### Official Review · Reviewer_EAug · 2026-03-19

**Soundness:** 3
**Presentation:** 3
**Significance:** 2
**Originality:** 2
**Overall Recommendation:** 4
**Confidence:** 3

**Summary:**

This paper tackles the setting of Mendelian randomization with potentially confounded instruments when data from multiple environments are available. The authors propose a representation learning framework that exploits multienvironment data to recover latent exogenous components of genetic instruments suitable for IV regression. The paper provides theoretical insights and synthetic and semi-synthetic experiments using data from the All of Us Biobank.

**Compliance With Llm Reviewing Policy:**

Affirmed.

**Final Justification:**

The rebuttal addressed my main concerns. I raised my score. However, I think the required assumptions are still pretty hard for practical application, and real-world insights would be interesting, thus I vote for weak accept.

**Key Questions For Authors:**

1. Could you elaborate more on the contribution of this work (e.g., compared to Yao et al (2024)), i.e. is there also a fundamental theoretical contribution or rather the application to MR?
2. Given that W (and maybe also V) are learned, does 2SLS not also assume linearity/structural knowledge about the SCMs? I think other flexible or non-parametric methods like DeepIV/KernelIV could be applied on top, but I think the IV estimation part should be discussed in more detail.
3. Empirically, HSIC and MMD losses are often hard to optimize in representation learning. How robust where the results against different kernels and lambdas here?
4. Can you provide full real-world results for some insights? Even some post hoc interpretability of the learned representations might be interesting to identify genetic variants that belong to valid instruments vs the ones that are confounded.

**Limitations:**

yes

**Strengths And Weaknesses:**

### Strengths:
- MR with unobserved confounding is a highly relevant research field and new methods for this setting (in this case with multiple environment data) are important for estimating the effects of treatments/exposures in healthcare.
- The paper is transparent with its assumptions regarding identification of the latent factors and provides a proper theoretical background.
- The paper provides synthetic and semi-synthetic experiments based on real-world data from healthcare using the All of Us database.

### Weaknesses:
- The assumptions are strong and hard to verify/understand. For example, Assumption 3 in Setting 2.1 seems pretty arbitrary (distributional invariance of the latent IV across environments). I think a discussion and intuition on when this actually holds/makes sense and whether this assumption can at least be backed up by expert information in certain cases, is necessary to evaluate the practical usefulness of the approach. Additionally, in Setting 2.1 the assumption that f(W,V) is invertible should also be discussed more regarding practicality, since a naive intuition is that in practice Z might lose some information regarding some unknown ground truth IV. I guess this is not necessarily a problem as long as the learned representation of W is still an IV (even though maybe with less predictive power of D), but this makes Assumption 3 even harder to verify for such an “ IV subset” W. So overall I think, examples and intuition when these assumptions are reasonable and when not would really strengthen the paper.
- The contribution is a bit unclear to me. It seems like the identification of latent variables in this manner is not new, however it is unclear to me if the application with multiple environments in the IV setting with unobserved confounding is novel, or whether this already existed before and only the transfer between IV to MR is the novelty.
- I think the identification of the causal effect even after the learned representations should be discussed a bit more into detail, i.e., the authors use 2SLS but does this not require some additional structural assumptions regarding the SCM additional to Defintion 2.1?
- Some full real-world experiments (even if benchmarking is not possible) would be really interesting for giving insights into differences in effect estimates of their method vs the ones using Z naively as an instrument (i.e. checking whether some existing real-world findings might actually be biased)

---

> ### Author Rebuttal · Authors · 2026-03-30
>
> We thank the reviewer for recognizing the relevance of this work and for their constructive feedback.
>
> ## Response to weakness
>
> ### W1: Assumptions in Setting 2.1
>
> **We discussed our main motivation in Section 1.1**, that genetic variants can be viewed as mixtures of invariant biological signals and environment-specific factors (Wang et al., 2020; 2023). **Another motivation follows from Proposition 2.3** (see after Remark 2.3): to obtain a valid instrument from observed $Z$, $Z$ must admit a decomposition into two independent components, one of which can serve as a valid instrument. **As the reviewer correctly pointed out, a non-injective mixing may cause $Z$ to lose information.** This means that lost information of $W$ cannot be recovered from $Z$, although the recoverable part may still help in MR. We are happy to clarify this in the revision. We will also connect our setup with Example 2.1. Specifically, that $V$ corresponds to $U_1$ and $W$ corresponds to $U_2$ and $U_3$.
>
> ### W2: Clarification of contributions
>
> Yes, **representation learning using multiple environments' data applied to the MR setting to specifically remove unobserved confounding is novel.** As mentioned in Section 1.1, prior works using representations for IV lack identifiability guarantees. We also extend identification theory by explicitly discussing the effect of estimated latent dimension, and show in experiments that misspecification can bias causal estimates (Figure 5). We will update the last paragraph in Section 1.1 to reflect this.
>
> ### W3: Discussion of causal effect identification and the 2SLS estimator
>
> **We discuss identification at the end of Section 2.2** that identification via (1) is characterized by a standard rank condition. This condition is well-known in the IV literature and is not altered by using learned representations as instruments.
>
> **Regarding estimation, consistency of 2SLS does not require a structurally linear instrument–exposure relationship**, only relevance in the linear projection sense (e.g., Newey and Powell, 2003). We focus on 2SLS for simplicity, but the learned representation can be used with other IV estimators depending on the target estimand and assumptions, including but not limited to PLSCMs.
>
> Reference:
>
> Newey, W. K., & Powell, J. L. (2003). Instrumental variable estimation of nonparametric models. Econometrica, 71(5), 1565-1578.
>
> **W4: Real-world experiments**
>
> We agree that analyzing biases in prior studies would be interesting. We did not include a full real-world experiment due to the lack of ground truth. As our goal is methodological (study how representation learning can support MR), we avoid making biological claims and instead use semi-synthetic experiments to compare observed and learned instruments. Applying our method to historical studies is of independent interest and beyond the scope of this paper.
>
> ## Response to questions
>
> ### Q1: Contribution compared to Yao et al (2024)
>
> While Yao et al. (2024) give a unifying perspective on latent variable identification using invariance constraints, **we showing that representation learning is a viable tool for deconfounding in MR**. In particular, **the ambiguity of learned representations (up to invertible transformations) does not hinder causal effect estimation in this setting**, which is not always the case as shown by Yao et al (2025). See also W2.
>
> Reference:
>
> Yao, D., Huang, S., Cadei, R., Zhang, K., & Locatello, F. (2025). The third pillar of causal analysis? A measurement perspective on causal representations. In Advances in Neural Information Processing Systems (NeurIPS 2025).
>
> ### Q2: Assumptions of 2SLS
>
> Please see W3.
>
> ### Q3: How robust were the results against different kernels and lambdas used in HSIC and MMD
>
> The lambdas in (8) are selected by cross-validation (see Section 5.2). Results of different invariance losses are given in Appendix E3. Results using different independence losses, including HSIC with a different kernel (degree-3 polynomial kernel) and a simple orthogonality, are given in **Tables R1a and R1b** below. Here, $Z$ is generated from a poly 3 mixing. For both estimators, HSIC and MMD using poly kernels of degree 2 or 3 all work well, while we see worse results when using the simple losses for invariance and independence, especially their combination.
>
> **Table R1a** $2\text{SLS}(\widehat{W})$
>
> | Ind. \ Inv.  | Mean-Var | MMD-Poly 2 | MMD-Poly 3 |
> |---|---|---|---|
> | Orth | 0.242 (0.011) | 0.106 (0.090) | 0.084 (0.025) |
> | HSIC-Poly 2 | 0.033 (0.078) | 0.012 (0.094) | 0.013 (0.049) |
> | HSIC-Poly 3 | 0.063 (0.184) | 0.038 (0.096) | 0.068 (0.084) |
>
> **Table R1b** $\text{PO}(\widehat{V})\text{-2SLS}(\widehat{W})$
>
> | ind. \ inv. | Mean-Var | Poly 2 | Poly 3 |
> |---|---|---|---|
> | Orth | 0.233 (0.011) | 0.135 (0.044) | 0.085 (0.023) |
> | Poly 2 | -0.004 (0.027) | -0.016 (0.026) | -0.003 (0.025) |
> | Poly 3 | -0.016 (0.035) | -0.009 (0.042) | -0.005 (0.024) |
>
> ### Q4: real-world experiment
>
> Please see W4.

---

> > ### Author Rebuttal · Reviewer_EAug · 2026-04-03
> >
> > The rebuttal addressed my main concerns. I raised my score. However, I think the required assumptions are still pretty hard for practical application, and real-world insights would be interesting, thus I vote for weak accept.

---

### Decision · Program_Chairs · 2026-04-30

**Decision:**

Accept (regular)

**Comment:**

This paper addresses the problem of invalid instruments in Mendelian Randomization (MR) by leveraging representation learning with multi‑environment data to extract invariant components from confounded instruments and use them as valid instruments for causal effect estimation. It presents a clear theoretical framework with sound identification results under explicit assumptions, which are shown to be sufficient for recovering valid instruments from confounded ones. Estimators derived from these theoretical insights are evaluated through synthetic and semi‑synthetic experiments, providing empirical evidence for the benefits of learning latent representations in this setting.

All reviewers commend the originality and potential usefulness of the paper’s contributions, though they differ somewhat in their assessment of their overall significance and novelty. Several concerns were also raised by the reviewers, including in particular the strength and practical plausibility of the identification assumptions (e.g., distributional invariance and invertibility), which may be easy to violate or difficult to verify in real MR settings; the limited intuition and concrete examples clarifying when these assumptions hold; the unclear degree of novelty relative to prior representation learning approaches; and gaps in the empirical evaluation, especially the absence of real data experiments.

Most of these concerns were carefully and adequately addressed in the authors’ rebuttal. While some reviewers continue to express reservations about the limited validation on fully real‑world applications, they also acknowledge the practical difficulty of obtaining real datasets with reliable ground truth, as noted by the authors.

My own assessment aligns with the general consensus that this is a strong paper, and I am pleased to recommend acceptance.